# The RIG-I-like receptor LGP2 inhibits Dicer-dependent processing of long double-stranded RNA and blocks RNA interference in mammalian cells

Annemarthe G van der Veen[1,*] (ID), Pierre V Maillard[1,†], Jan Marten Schmidt[1,‡], Sonia A Lee[1], Safia Deddouche-Grass[1,§], Annabel Borg[2], Svend Kjær[2], Ambrosius P Snijders[3] & Caetano Reis e Sousa[1,**] (ID)

## Abstract

In vertebrates, the presence of viral RNA in the cytosol is sensed by members of the RIG-I-like receptor (RLR) family, which signal to induce production of type I interferons (IFN). These key antiviral cytokines act in a paracrine and autocrine manner to induce hundreds of interferon-stimulated genes (ISGs), whose protein products restrict viral entry, replication and budding. ISGs include the RLRs themselves: RIG-I, MDA5 and, the least-studied family member, LGP2. In contrast, the IFN system is absent in plants and invertebrates, which defend themselves from viral intruders using RNA interference (RNAi). In RNAi, the endoribonuclease Dicer cleaves virus-derived double-stranded RNA (dsRNA) into small interfering RNAs (siRNAs) that target complementary viral RNA for cleavage. Interestingly, the RNAi machinery is conserved in mammals, and we have recently demonstrated that it is able to participate in mammalian antiviral defence in conditions in which the IFN system is suppressed. In contrast, when the IFN system is active, one or more ISGs act to mask or suppress antiviral RNAi. Here, we demonstrate that LGP2 constitutes one of the ISGs that can inhibit antiviral RNAi in mammals. We show that LGP2 associates with Dicer and inhibits cleavage of dsRNA into siRNAs both in vitro and in cells. Further, we show that in differentiated cells lacking components of the IFN response, ectopic expression of LGP2 interferes with RNAi-dependent suppression of gene expression. Conversely, genetic loss of LGP2 uncovers dsRNA-mediated RNAi albeit less strongly than complete loss of the IFN system. Thus, the inefficiency of RNAi as a mechanism of antiviral defence in mammalian somatic cells can be in part attributed to Dicer inhibition by LGP2 induced by type I IFNs. LGP2-mediated antagonism of dsRNA-mediated RNAi may help ensure that viral dsRNA substrates are preserved in order to serve as targets of antiviral ISG proteins.

**Keywords** double-stranded RNA; innate immunity; RIG-I-like receptor family; RNA interference; viral infection
**Subject Categories** Immunology; RNA Biology
**The EMBO Journal (2018) 37: e97479**

## Introduction

In all kingdoms of life, organisms are under constant threat from viral pathogens and effective strategies to combat viral infection are crucial for survival. To this end, all species have developed cell-intrinsic and cell-extrinsic mechanisms of antiviral defence. In mammals, the type I interferon (IFN) response constitutes an effective means to halt viral intruders. This innate immune response is initiated by the recognition of virally derived nucleic acids within the host cells by a dedicated group of receptors, which signal to induce a transcriptional response resulting in the production and secretion of type I IFNs (mainly IFN-α and IFN-β; Goubau *et al*, 2013; Schneider *et al*, 2014; Wu & Chen, 2014; Schlee & Hartmann, 2016). These key antiviral cytokines signal in an autocrine and paracrine fashion via the type I interferon receptor (IFNAR) to induce the expression of hundreds of interferon-stimulated genes (ISGs) that encode proteins that inhibit viral replication and dissemination (Goubau *et al*, 2013; Schneider *et al*, 2014).

Central to the detection of virally derived ribonucleic acids in the cytosol of infected cells are the RIG-I-like receptors (RLRs). This family of DExH/D helicases encompasses RIG-I (retinoic acid-inducible gene I), MDA5 (melanoma differentiation-associated gene 5) and LGP2 (laboratory of genetics and physiology 2; Goubau *et al*, 2013; Wu & Chen, 2014; Schlee & Hartmann, 2016). RIG-I and MDA5 share a similar domain organisation. Recognition of an RNA substrate by the C-terminal domain (CTD) of RIG-I or MDA5 leads

1 Immunobiology Laboratory, The Francis Crick Institute, London, UK
2 Structural Biology Platform, The Francis Crick Institute, London, UK
3 Mass Spectrometry Platform, The Francis Crick Institute, London, UK
*Corresponding author. Tel: +44 20 3796 1344; E-mail: annemarthe.vanderveen@crick.ac.uk
**Corresponding author. Tel: +44 20 3796 1310; E-mail: caetano@crick.ac.uk
†Present Address: Division of Infection and Immunity, University College London, London, UK
‡Present address: Friedrich Miescher Institute for Biomedical Research, Basel, Switzerland
§Present address: Open Innovation Access Platform, Sanofi Strasbourg, Strasbourg, France

to structural rearrangements that are transmitted via the conserved helicase domain and allow the two N-terminal caspase activation and recruitment domains (CARDs) to trigger oligomerisation of the adaptor MAVS (mitochondrial antiviral signalling) on the mitochondrial membrane, which in turn leads to IRF-3, IRF-7 and NF-κB activation, nuclear translocation and IFN gene transcription (Goubau *et al*, 2013; Wu & Chen, 2014; Sohn & Hur, 2016). Whilst RIG-I recognises 5′di- or tri-phosphates at the base-paired extremities of viral RNA, MDA5 and LGP2 recognise long double-stranded RNA (dsRNA), the precise features of which are less well-defined (Hornung *et al*, 2006; Kato *et al*, 2006; Pichlmair *et al*, 2006, 2009; Goubau *et al*, 2014). LGP2 remains the most enigmatic member of the RLR family. Whilst it shares the conserved helicase domain and CTD with RIG-I and MDA5, the lack of the N-terminal CARD domains renders it signalling-incompetent. Instead, LGP2 potentiates signalling via MDA5 by increasing the rate of MDA5–RNA interactions and by promoting nucleation of MDA5 filaments on dsRNA (Bruns *et al*, 2014; Bruns & Horvath, 2015). Loss of LGP2 therefore leads to increased sensitivity to viruses detected by MDA5 such as picornaviridae (Venkataraman *et al*, 2007; Satoh *et al*, 2010). LGP2 may also impact RIG-I signalling (Rothenfusser *et al*, 2005; Yoneyama *et al*, 2005), and alternative functions have been suggested by the fact that LGP2-deficient mice have impaired CD8[+] T-cell responses to West Nile virus or lymphocytic choriomeningitis virus that appear unrelated to IFN production (Suthar *et al*, 2012).

In contrast to the protein-based type I IFN system found in vertebrates, antiviral immunity in plants and invertebrates relies on an RNA-based, sequence-specific defence pathway known as RNA interference (RNAi; Ding & Voinnet, 2007; Kemp & Imler, 2009; Wilson & Doudna, 2013; tenOever, 2016). The importance of RNAi in antiviral defence in these organisms is underscored by the evolution in most plant and insect viruses of viral-encoded RNAi antagonists, known as viral suppressors of RNA silencing (VSRs) that block various steps in the RNAi pathway (Ding & Voinnet, 2007; Wu *et al*, 2010). In the absence of VSRs, viral dsRNA generated during viral infection and replication is rapidly cleaved by the type III ribonuclease Dicer into virus-derived small interfering RNAs (viRNAs), which are loaded onto an Argonaute (Ago) family protein to form the RNA-induced silencing complex (RISC) that targets complementary viral RNA for cleavage (Ding & Voinnet, 2007; Kemp & Imler, 2009; Wilson & Doudna, 2013; tenOever, 2016). Interestingly, the RLR family and Dicer share the conserved DExD/H helicase domain and have comparable substrate specificity (dsRNA) although their respective functions (signalling versus catalytic processing) are distinct (MacKay *et al*, 2014; Paro *et al*, 2015).

All components of the RNAi pathway are conserved in mammals, and siRNA-based gene silencing is frequently exploited as an experimental tool in mammalian cells. However, Dicer's catalytic function appears mainly confined to precursor microRNA (pre-miRNA) processing, and its antiviral capacity in mammals remains much debated (Parameswaran *et al*, 2010; Cullen *et al*, 2013; Li *et al*, 2013, 2016; Maillard *et al*, 2013; Backes *et al*, 2014; Ding & Voinnet, 2014; Kennedy *et al*, 2015; Jeffrey *et al*, 2017; tenOever, 2017). Some studies have reported that RNAi can impact antiviral immunity in mammals during influenza A virus, hepatitis C virus, Nodamura virus and, more recently, human enterovirus 71 infection (Wang *et al*, 2006; Matskevich & Moelling, 2007; Li *et al*, 2013,

2016; Maillard *et al*, 2013; Qiu *et al*, 2017). In contrast, others have found low abundance of viRNAs in mammalian somatic cells infected with various viruses, and only a modest effect of Dicer deficiency on viral replication, suggesting that RNAi is not an active mechanism of antiviral defence in most mammalian cell types (Parameswaran *et al*, 2010; Girardi *et al*, 2013; Backes *et al*, 2014; Bogerd *et al*, 2014; Schuster *et al*, 2017). RNAi may be particularly important in undifferentiated mammalian cells, and clear evidence of the existence of endogenous RNAi, dsRNA-mediated RNAi (dsRNAi), or antiviral RNAi has been documented in oocytes, embryonic teratocarcinoma cell lines and mouse embryonic stem cells (mESCs), respectively (Billy *et al*, 2001; Flemr *et al*, 2013; Maillard *et al*, 2013). In mESCs, viral infection leads to the Dicer-dependent emergence of viRNAs that associate with Ago2 and inhibit replication of a VSR-deficient homologous virus (Maillard *et al*, 2013). Those observations led to the hypothesis that the IFN system, which is attenuated in undifferentiated cells and germ cells (Pare & Sullivan, 2014), may inhibit or mask antiviral RNAi in mammalian somatic cells. In line with this hypothesis, we recently demonstrated that ablation of the IFN response by genetic removal of IFNAR or MAVS in mouse embryonic fibroblasts (MEFs) enables dsRNAi (Maillard *et al*, 2016). Our study, therefore, indicated that the IFN system, through induction of one or more ISGs, suppresses RNAi. This was likely happening at the level of Dicer processing, as the response to exogenous siRNAs was unaltered in IFN-deficient versus IFN-sufficient cells (Maillard *et al*, 2016).

Here, we provide further evidence for this notion and demonstrate that LGP2 constitutes one ISG that inhibits RNAi in mammalian somatic cells. Using a proteomics-based approach, we identify LGP2 as a Dicer interactor. We show that LGP2, but not MDA5 or RIG-I, interferes with Dicer processing of dsRNA substrates both *in vitro* and *in vivo*. Consequently, the ectopic expression of LGP2 blocks dsRNAi in IFNAR-deficient cells, whilst genetic ablation of LGP2 uncovers dsRNAi in MEFs. Together, these findings shed light on the role of LGP2 and indicate that expression of ISGs actively suppresses the antiviral RNAi response in IFN-competent mammalian somatic cells through inhibition of Dicer processing of dsRNA.

# Results

## A proteomics approach identifies an interaction between LGP2 and the Dicer complex

To better understand the functions of the RNA helicase LGP2, we used proteomics to identify binding partners. We transiently expressed a plasmid encoding FLAG-tagged human LGP2 or a control plasmid in HEK293 embryonic kidney fibroblasts and treated these with recombinant type I IFN or lipofectamine-complexed poly(I:C), a dsRNA analogue, in order to mimic a state of viral infection (Fig 1A). LGP2 complexes were then retrieved by FLAG immunoprecipitation (IP) and isolated from the resin by peptide elution (Fig EV1A). Following concentration, the complexes were subjected to SDS–PAGE, excised and further processed by trypsin digestion for LC-MS/MS analysis (Fig EV1B). Amongst the strongest LGP2 interactors in both the type I IFN and poly(I:C)-treated samples, we identified the endoribonuclease Dicer, as well as

several known co-factors of Dicer, including TRBP (HIV TAR RNA-binding protein), PACT (protein activator of PKR) and PKR (protein kinase R; Fig 1B), all of which appear at negligible frequency in the "Contaminant Repository for Affinity Purification" databases for FLAG IPs (Mellacheruvu *et al*, 2013).

We verified the interaction between Dicer and LGP2 by co-IP and immunoblotting. We expressed FLAG-LGP2 in HEK293 cells and left the cells untreated or treated them with type I IFN, uncomplexed or lipofectamine-complexed poly(I:C). Interaction between LGP2 and Dicer was observed in all cases and further enhanced in the presence of cytosolic dsRNA (Fig 1C). To determine the impact of viral infection on the stability of the LGP2–Dicer complex, we expressed FLAG-LGP2 in HEK293 cells and infected these with the single-stranded positive-strand RNA virus encephalomyocarditis virus (EMCV), a member of the picornaviridae that generates long dsRNA products that trigger type I IFN induction via MDA5 (Pichlmair *et al*, 2009). As observed upon poly(I:C) treatment, the presence of EMCV-derived viral dsRNA modestly increased the interaction between LGP2 and Dicer, possibly via their shared ability to bind dsRNA (Fig 1D). We subsequently tested whether the ability to bind Dicer was unique to LGP2 or a common feature of other members of the family of RLRs. Only FLAG-LGP2, but not FLAG-tagged MDA5, RIG-I, or an unrelated control protein (p97) was able to appreciably interact with Dicer (Fig 1E). We next tested whether LGP2 is also associated with other components of the Dicer complex, PACT and PKR, which were also identified by mass spectrometry as LGP2 interactors. Both PACT and PKR co-purified with LGP2, as did epitope-tagged Ago2, another RNAi machinery component (Fig 1F). Despite the identification of TRBP by mass spectrometry as another LGP2 interactor, we failed to verify binding between LGP2 and TRBP in co-IP experiments, possibly due to the quality of the antibodies used (data not shown).

To dissect the contribution of RNA molecules to the observed interactions, we tested whether the latter are altered upon inclusion of ribonucleases during co-IP. Addition of the promiscuous single-stranded RNA ribonucleases RNase A and RNase I weakened but did not abolish the interaction between LGP2 and Dicer (Fig 1F). Other interactions were unaffected (LGP2-Ago2, LGP2-PKR) or appeared more stable (LGP2-PACT; Fig 1F) despite efficient RNA cleavage by the enzymes as monitored by measuring RNA integrity in parallel experiments (data not shown). Although *bona fide* dsRNA does not naturally occur in uninfected mammalian cells, base-paired RNA formed by intramolecular base-pairing is present. Therefore, we also performed a co-IP in the presence of the base-paired RNA-specific ribonuclease RNase III (Fig 1G). Inclusion of RNase III did not affect the LGP2–Dicer interaction at steady state, although the increased stability observed in the presence of cytosolic dsRNA was abolished. We conclude from these experiments that LGP2 but not the related RNA sensors RIG-I and MDA5 interacts with Dicer and several of its co-factors at steady state, and in a manner that likely involves RNA-induced structural alterations to either protein.

## LGP2 interacts with Dicer via its C-terminal domain

To map the domains of LGP2 that bind to Dicer, we generated different truncations mutants (Fig 2A). Full-length LGP2 is composed of a conserved N-terminal DExH helicase domain (NTD), which contains the ATPase domain, a pincer (P) domain containing several coiled-coil motifs, and a C-terminal domain (CTD), which is conserved amongst the RIG-I-like helicases and essential for RNA binding (Goubau *et al*, 2013; Bruns & Horvath, 2015). The FLAG-tagged CTD of LGP2 was able to co-IP Dicer, unlike two different NTD fragments (Fig 2B). Mutation of a conserved lysine residue near the C-terminus of LGP2 (K634), which has previously been shown to prevent RNA binding (Li *et al*, 2009; Pippig *et al*, 2009), reduced the association of the CTD of LGP2 with Dicer (Fig 2C). However, mutation of the same residue within the context of the full-length protein weakened but did not abolish binding to Dicer, suggesting that the N-terminal helicase domain, although by itself unable to bind Dicer, stabilises the interaction between the CTD and Dicer.

## Dicer does not impact RNA sensing and type I IFN signalling via the LGP2/MDA5 pathway

To determine the functional consequences of the identified interaction between LGP2 and the Dicer complex, we first tested the impact of the latter on RNA sensing via the MDA5/LGP2 pathway. To this end, we used siRNAs to deplete Dicer, PACT, TRBP or, as a positive control, MDA5, in HEK293 cells stably expressing FLAG-LGP2 (Fig 3A, left panel). Following siRNA-mediated knockdown of the indicated proteins, the cells were transfected with a reporter plasmid expressing the firefly luciferase gene under the control of the IFN-β promotor, followed by transfection with increasing doses of RNA extracted from EMCV-infected HeLa cells, a selective MDA5 agonist (Kato *et al*, 2006; Pichlmair *et al*, 2009). In contrast to MDA5 knockdown, which abrogated type I IFN signalling, we failed to observe a marked effect of Dicer, PACT or TRBP depletion on type I IFN production, as measured by luciferase activity (Fig 3A, right panel). Note that a recent study reported a modest effect of TRBP overexpression on LGP2/MDA5-dependent IFN production triggered by EMCV in a similar reporter assay (Komuro *et al*, 2016). In contrast to overexpression (Komuro *et al*, 2016), TRBP depletion did not affect IFN production in our hands. We subsequently used an RNA IP approach to investigate whether Dicer alters the binding of LGP2 to viral stimulatory RNA (Fig 3B, left panel). FLAG-LGP2-expressing HeLa cells were treated with a non-targeting control siRNA or a siRNA targeting Dicer and subsequently infected with EMCV (MOI 1) for 15 h (Fig EV2A). Following cell lysis and IP using a FLAG or a control IgG antibody (Fig EV2B), RNA was extracted from the immunoprecipitates and transfected into HEK293 IFN-β reporter cells. Whilst IFN-stimulatory RNA was strongly enriched in the LGP2 IPs compared to the control IgG IPs, as reported (Deddouche *et al*, 2014), this was not altered by depletion of Dicer (Fig 3B, right panel). We conclude that Dicer does not appreciably affect RNA sensing and type I IFN production via activation of the LGP2/MDA5 pathway.

## LGP2 inhibits Dicer-dependent processing of long dsRNA *in vitro*

As we did not find evidence for a role for Dicer in LGP2/MDA5 signalling, we next assessed whether LGP2 might impact Dicer function. The endoribonuclease activity of Dicer can process two types of substrates: pre-miRNAs and dsRNA. To test whether LGP2 can alter

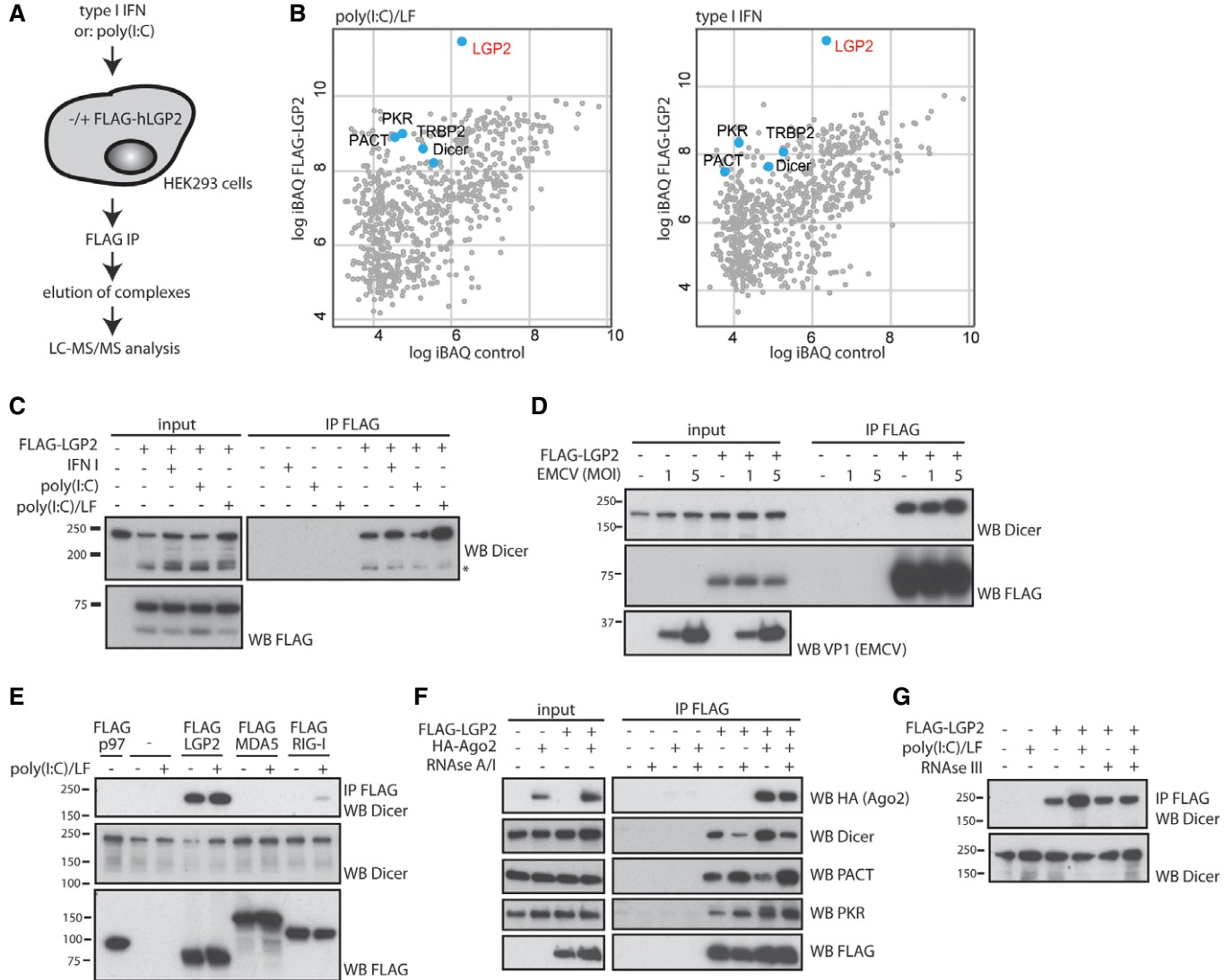

**Figure 1. A mass spectrometry-based approach reveals an association between the RNA helicase LGP2 and the endoribonuclease Dicer and its co-factors.**

A  Schematic illustration of the experimental approach used to identify LGP2 interactors. HEK293 cells were transfected with a FLAG-human LGP2 expression plasmid or a control plasmid for 24 h, transfected with 1 μg/ml poly(I:C) for 4 h or treated with type I IFN (200 U/ml) for 20 h and subsequently lysed and subjected to FLAG immunoprecipitation. FLAG-LGP2 complexes were eluted from the resin using 3FLAG-peptide, concentrated, separated by SDS–PAGE, excised and analysed by LC-MS/MS.

B  Graphs display the intensity-based absolute quantification (iBAQ) score of proteins identified in the LGP2 IP versus the control IP of cells treated with Lipofectamine (LF)-complexed poly(I:C) or type I IFN. Dicer, as well as its co-factors TRBP, PACT and PKR, is highlighted.

C  HEK293 cells were transfected with FLAG-LGP2 and treated with type I IFN, uncomplexed poly(I:C) or Lipofectamine (LF)-complexed poly(I:C) as described in (A). Cells were lysed and subjected to FLAG immunoprecipitation, followed by SDS–PAGE and immunoblotting using the indicated antibodies. The asterisk indicates a degradation product of Dicer that we occasionally observe.

D  HEK293 cells were transfected with FLAG-LGP2 for 24 h followed by infection with EMCV at the indicated MOI for 6 h. Cells were processed as in (C).

E  HEK293 cells were transfected with FLAG-LGP2, FLAG-MDA5, FLAG-RIG-I or a non-related control protein FLAG-p97 and processed as in (C and D).

F  FLAG-LGP2 was co-transfected with HA-tagged Argonaute 2 (HA-Ago2) in HEK293 cells and subjected to FLAG immunoprecipitation in the presence or absence of RNase A and RNase I. Immunoprecipitates were analysed by SDS–PAGE and immunoblotting using the indicated antibodies.

G  FLAG-LGP2 was transfected in HEK293 cells and 24 h later, cells were transfected with poly(I:C) for 4 h. FLAG immunoprecipitation was performed in the absence or presence of RNase III, and the samples were subsequently analysed by SDS–PAGE and immunoblotting with the indicated antibodies.

Source data are available online for this figure.

the Dicer-dependent processing of pre-miRNAs into mature miRNAs, we set up an *in vitro* dicing assay by combining recombinant immunopurified FLAG-tagged human Dicer (isolated from HEK293T

cells, Fig EV3A) with synthetic pre-Let7a, a highly conserved and abundant miRNA. The pre-Let7a harboured two Cy5-labelled nucleotides within the stem of the mature microRNA structure, allowing us

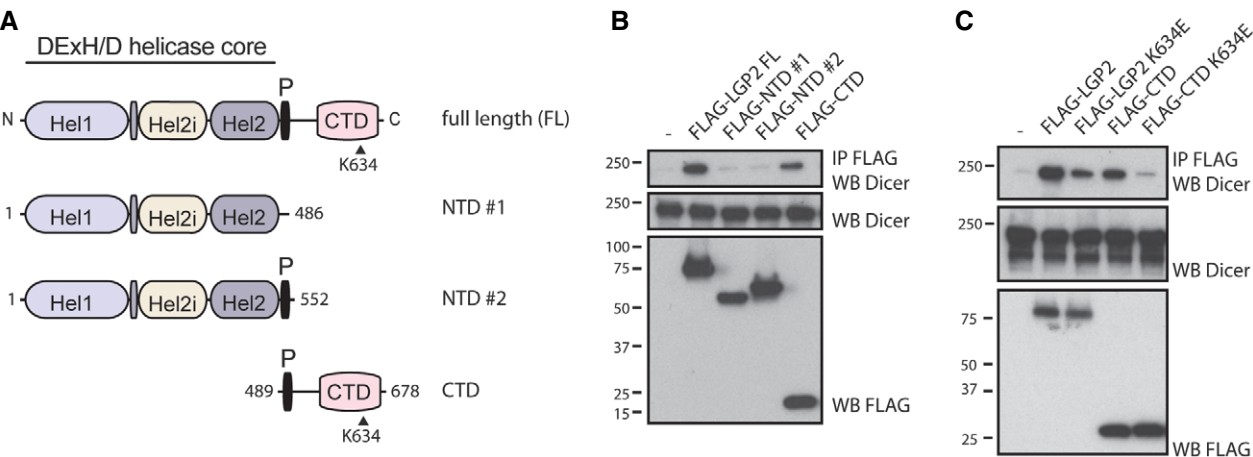

**Figure 2.  Interaction with Dicer is mediated by the C-terminal domain of LGP2 and enhanced by substrate engagement and the helicase domain.**

A   Schematic illustration of the domain structure of LGP2. LGP2 contains a conserved DExH/D helicase domain, subdivided into the helicase 1 domain (Hel1), the helicase insertion domain (Hel2i) and the helicase 2 domain (Hel2). The Pincer motif (P) is a coiled-coil domain, and the C-terminal domain (CTD) is important for RNA binding. Further denoted is lysine 634, a residue critical for RNA binding.

B   The C-terminal domain of LGP2 is critical for binding to Dicer. The indicated FLAG-tagged constructs were transfected in HEK293 cells, followed by FLAG immunoprecipitation, SDS–PAGE and immunoblotting using the indicated antibodies.

C   Lysine 634 of LGP2 is important for the LGP2–Dicer interaction. The indicated FLAG-tagged constructs and mutants were transfected in HEK293 cells and processed as in (B).

Source data are available online for this figure.

to monitor processing into mature Let7a by in-gel fluorescence after electrophoresis in a denaturing polyacrylamide gel. Pre-Let7a was cleaved highly efficiently into mature Let7a by wild-type recombinant Dicer but not a Dicer mutant in which the catalytic sites have been mutated (D1320A/D1709A; Fig 4A). Using this assay, we found that pre-miRNA processing by Dicer was not inhibited in the presence of FLAG-tagged human LGP2 (purified from insect cells) across a large dose range (Fig 4A). We next examined the effect of LGP2 on the Dicer-dependent processing of dsRNA into siRNAs. We set up a dicing assay using *in vitro*-synthesised long dsRNA corresponding to the first 200 nucleotides of the Renilla luciferase coding sequence (dsRNA-RL) that we internally labelled with Cy5 by incorporation of Cy5-cytosine during *in vitro* transcription. As expected, Cy5-labelled dsRNA-RL was processed into siRNAs when combined with wild-type Dicer but not a catalytic mutant (Fig EV3B). Notably, in the presence of LGP2, cleavage of dsRNA into siRNAs by Dicer was reduced approximately threefold, as evidenced by an accumulation of the dsRNA substrate and a decrease in small RNA production (Fig 4B). Importantly, the inhibitory effect was unique to LGP2, as inclusion of FLAG-tagged MDA5 or FLAG-tagged RIG-I had little effect on dicing efficiency (Figs 4C and EV3C). In contrast to full-length LGP2, purified LGP2 CTD only modestly decreased *in vitro* dicing of dsRNA despite its ability to associate with Dicer and to bind dsRNA with high affinity (Li *et al*, 2009; Pippig *et al*, 2009; Figs 4D and EV3D). Furthermore, there was little difference between the LGP2 CTD and a CTD K634E mutant, which completely lacks RNA binding. Together, these results suggest that the inhibitory effect of LGP2 on dsRNA processing cannot be attributable to competition with Dicer for the dsRNA substrate (Fig 4D). We conclude that full-length LGP2, but not RIG-I or MDA5, specifically inhibits processing of dsRNA but not pre-miRNA by Dicer *in vitro*.

## Expression of LGP2 is sufficient to inhibit dsRNA processing by Dicer in cells lacking a type I IFN response

Having identified an inhibitory role for LGP2 on Dicer-dependent dsRNA processing using purified components, we next tested whether LGP2 has a similar function in mammalian cells. We used *Ifnar1*$^{-/-}$ MEFs that are unable to respond to type I IFN or upregulate ISGs and that we previously showed to be able to process dsRNA into siRNAs (Maillard *et al*, 2016). As LGP2 is expressed at very low levels in the absence of IFN receptor signalling (Fig EV4A), we used a lentiviral-based inducible system to express FLAG-LGP2 in a doxycycline-regulated manner in *Ifnar1*$^{-/-}$ cells, mimicking IFN-induced expression of LGP2 (hereafter referred to as *Ifnar1*$^{-/-}$ iLGP2 cells). We selected several single-cell clones in which doxycycline-induced LGP2 expression was verified by anti-FLAG immunoblotting (Fig 5A) and by intracellular staining and flow cytometry (Fig EV4B). We subsequently treated three of these clones with or without doxy-cycline for 24 h followed by transfection with long dsRNA encoding the first 200 nt of GFP (dsRNA-GFP). Twenty-four hours post-transfection, we extracted total RNA for Northern blot analysis using a probe specific for dsRNA-GFP. As reported (Maillard *et al*, 2016), dsRNA-GFP was cleaved into siRNAs of ~22 nt in length in *Ifnar1*$^{-/-}$ cells. Notably, we observed a dramatic reduction in small RNA production upon doxycycline-induced LGP2 expression (Fig 5B), but not upon induction of expression of the LGP2 CTD or the CTD K634E mutant (Fig EV4C–E). In contrast, LGP2 expression did not affect the production or processing of pre-miR16 into mature miR-16 (Fig 5B). We conclude that expression of LGP2 is sufficient to block dsRNA processing by Dicer in mammalian cells.

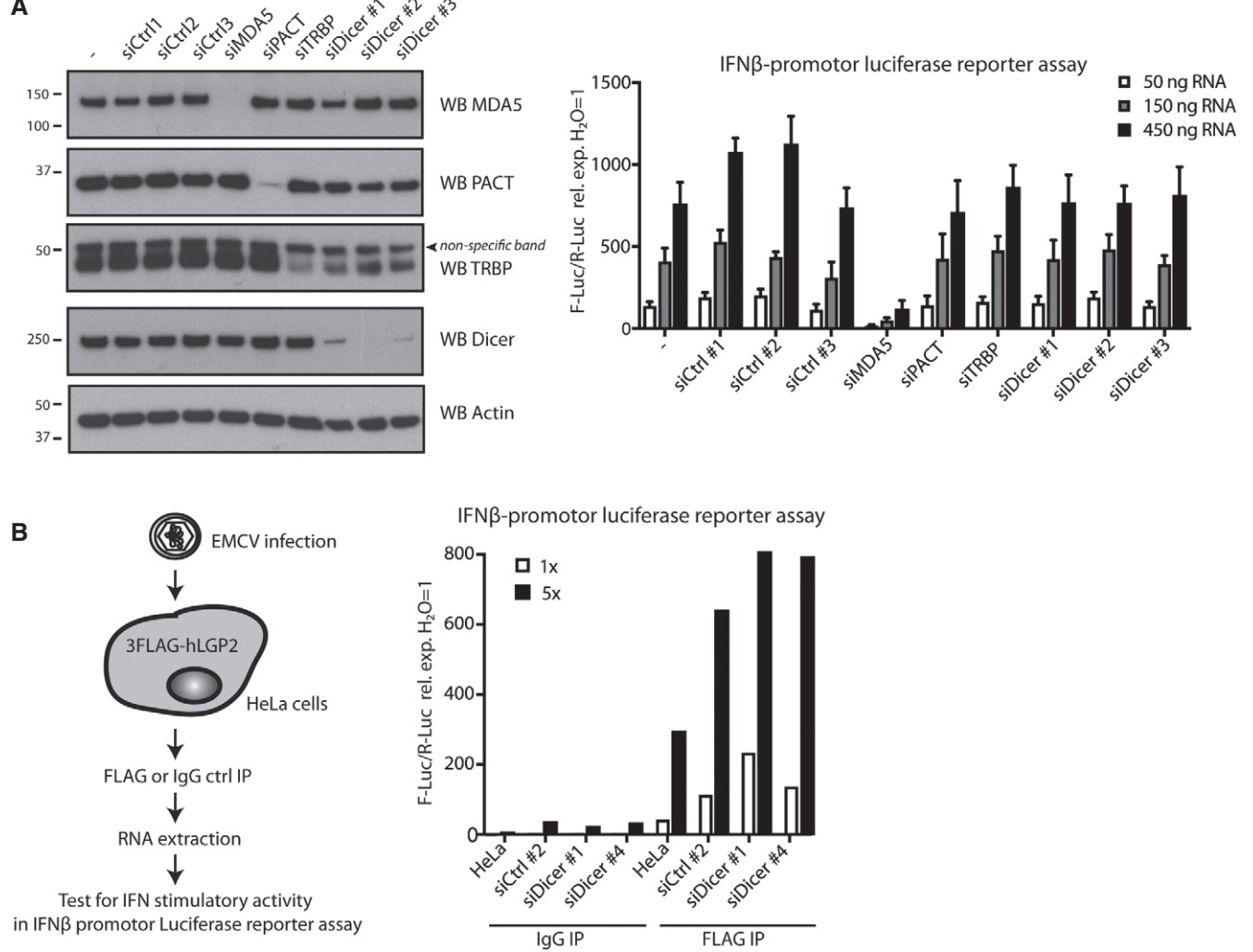

**Figure 3.  Dicer and its co-factors do not impact on the ability of LGP2 to bind viral RNA and induce MDA5-dependent type I IFN signalling.**

A  HEK293 cells stably expressing FLAG-LGP2 were treated with siRNAs targeting MDA5, TRBP, PACT or Dicer or non-targeting control siRNAs. Knockdown efficiency was determined by SDS–PAGE and immunoblotting using the indicated antibodies (left panel). β-Actin serves as loading control. Three days post-transfection, the IFN response to the indicated doses of total RNA extracted from EMCV-infected HeLa cells was tested in an IFN-β promotor luciferase-based reporter assay (right panel). Mean values and SEM of three independent experiments are shown. Statistical analysis was performed using one-way ANOVA with Sidak's multiple comparisons test as post-test for pairwise comparisons using untransfected cells as control for each dose. Significant differences were observed for siMDA5 only (*$P < 0.05$).

B  HeLa cells were treated with non-targeting control siRNAs or siRNAs targeting Dicer, transfected with FLAG-LGP2, and 48 h later infected with EMCV (MOI 1). Twelve hours post-infection, cells were lysed and LGP2 was retrieved by FLAG immunoprecipitation. The associated RNA was extracted from the immunoprecipitates and tested in the IFN-β promotor luciferase reporter assay in HEK293 cells at two doses (1× and 5×).

Source data are available online for this figure.

## LGP2 inhibits dsRNA-mediated RNA interference in mammalian cells

To determine the functional implications of the LGP2-dependent block of dsRNA processing by Dicer, we used a previously established flow cytometry-based dsRNAi reporter system that allows us to measure long dsRNA-mediated gene silencing at a single-cell level (Maillard *et al*, 2016). The *Ifnar1*$^{-/-}$ iLGP2 cells described above stably express a destabilised form of GFP (d2GFP), allowing us to monitor GFP expression levels by flow cytometry following transfection with Cy5-labelled dsRNA-GFP or, as a control, Cy5-dsRNA-RL.

Delivery of dsRNA-GFP, but not dsRNA-RL, induces a decrease in the expression of d2GFP in a Dicer- and Ago2-dependent manner in these MEFs (Maillard *et al*, 2016). We transfected four individual *Ifnar1*$^{-/-}$ iLGP2 clones, cultured in the absence or presence of doxycycline, with dsRNA-RL or dsRNA-GFP and monitored d2GFP expression 48 h later. In line with our previous observations, dsRNA-GFP delivery reduced d2GFP expression in live, Cy5-positive (transfected) *Ifnar1*$^{-/-}$ MEFs in the absence of doxycycline treatment. Importantly, doxycycline-induced expression of LGP2 blocked the dsRNAi-mediated inhibition of the d2GFP reporter in all four clones tested (Fig 6A). In contrast, doxycycline had no effect on

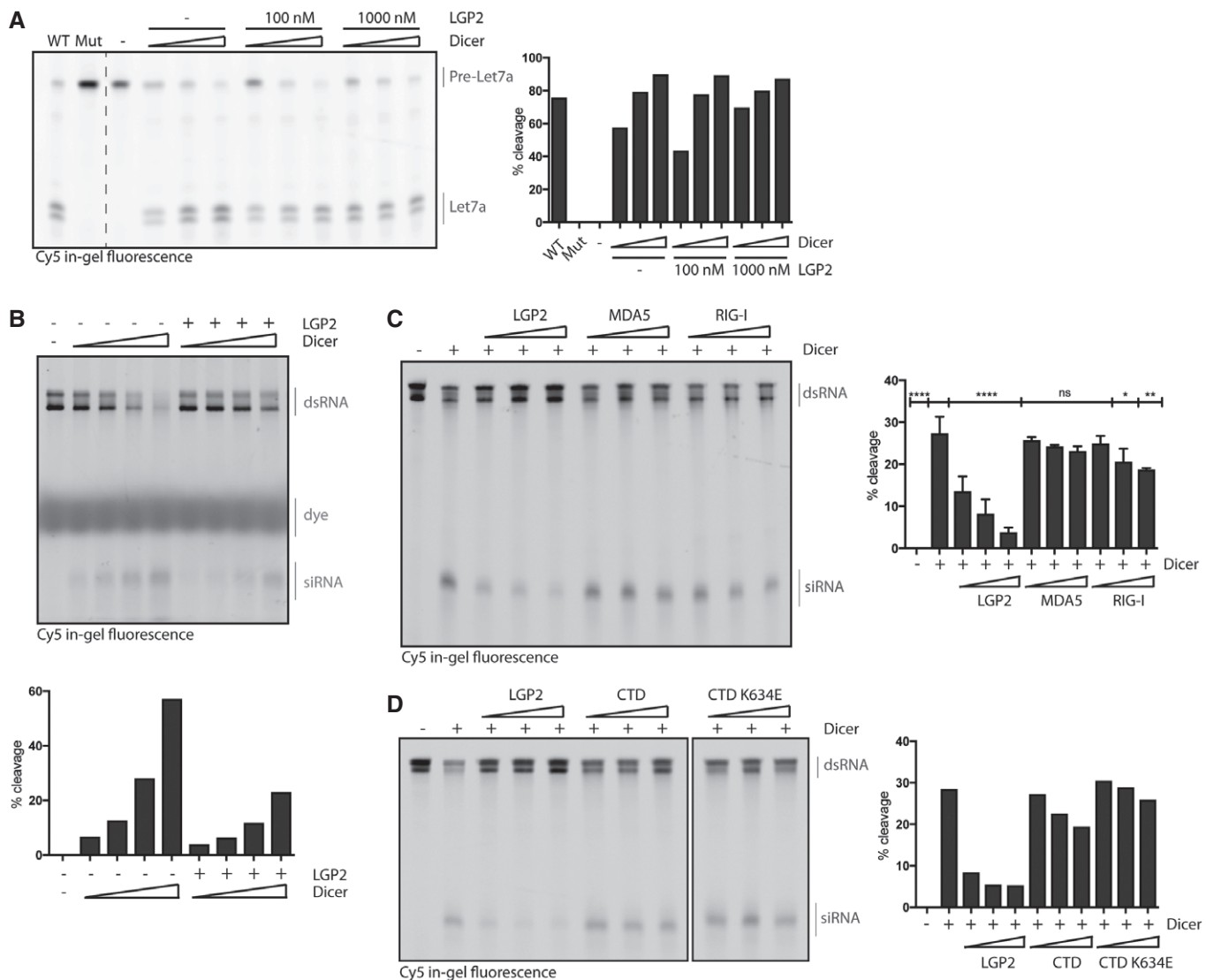

**Figure 4.  LGP2 but not MDA5 or RIG-I inhibits the processing of dsRNA by Dicer *in vitro*.**

A  Synthetic Cy5-labelled pre-Let7a (100 nM) was subjected to cleavage by immunopurified FLAG-tagged human Dicer (25/50/100 nM) in the absence or presence of recombinant human LGP2 for 1 h at 37°C. The reactions were analysed on a denaturing polyacrylamide gel by Cy5 in-gel fluorescence (left panel). Left lanes: pre-Let7a is processed by wild-type Dicer but not by a catalytic mutant (D1320A/D1709A). The bar graph depicts the percentage of cleavage, that is the signal intensity of small RNA relative to total RNA (right panel).

B  Increasing amounts of FLAG-Dicer (150/300/600/1,200 nM) were incubated with *in vitro*-synthesised 200-nt-long dsRNA internally labelled with Cy5 (200 nM), in the presence or absence of FLAG-LGP2 (500 nM) for 1 h at 37°C. The reactions were processed as in (A).

C  Cy5-labelled dsRNA (50 nM) was incubated with FLAG-Dicer (500 nM) and increasing amounts of FLAG-tagged LGP2, MDA5 or RIG-I (250/500/1,000 nM). The reactions were processed as in (A). Mean values and SD of three independent experiments are shown. Statistical analysis was performed using one-way ANOVA with Sidak's multiple comparisons test as post-test for pairwise comparisons using the second bar (dsRNA + Dicer) as a control (ns, not significant; *$P < 0.05$; **$P < 0.01$, ****$P < 0.0001$).

D  As in (C), using increasing amounts of FLAG-tagged full-length LGP2, CTD or CTD K634E (250/500/1,000 nM).

dsRNAi in the parental *Ifnar1*$^{-/-}$ MEFs that lacked iLGP2 expression constructs (Fig EV5A). Likewise, doxycycline-induced expression of FLAG-CTD or FLAG-CTD K634E did not appreciably impact dsRNAi in a consistent manner (Fig 6B). Similarly, dsRNAi was not affected by inducible expression of FLAG-RIG-I or FLAG-MDA5 (Fig EV5B–D), confirming the observations *in vitro* (Fig 4C and D) that full-length LGP2 but not the other members of the RLR family have the ability to block Dicer. Of note, doxycycline-inducible expression of LGP2 did not block dsRNAi in an embryonic carcinoma cell line (F9) that was

previously shown to efficiently process dsRNA into siRNAs whilst being refractory to activation of the IFN response (Fig EV6A and B; Billy *et al*, 2001). This observation indicates that LGP2 may work in concert with other ISGs or IFN-controlled mechanisms absent in F9 cells to suppress dsRNAi. Finally, to assess whether loss-of-function of LGP2 is sufficient to reveal dsRNAi, we generated MEFs from *Dhx58*$^{+/+}$ and *Dhx58*$^{+/-}$ (LGP2-sufficient) or *Dhx58*$^{-/-}$ (LGP2-deficient) littermate embryos and transduced these with the lentiviral d2GFP reporter. A reduction in GFP expression was seen in *Dhx58*$^{-/-}$

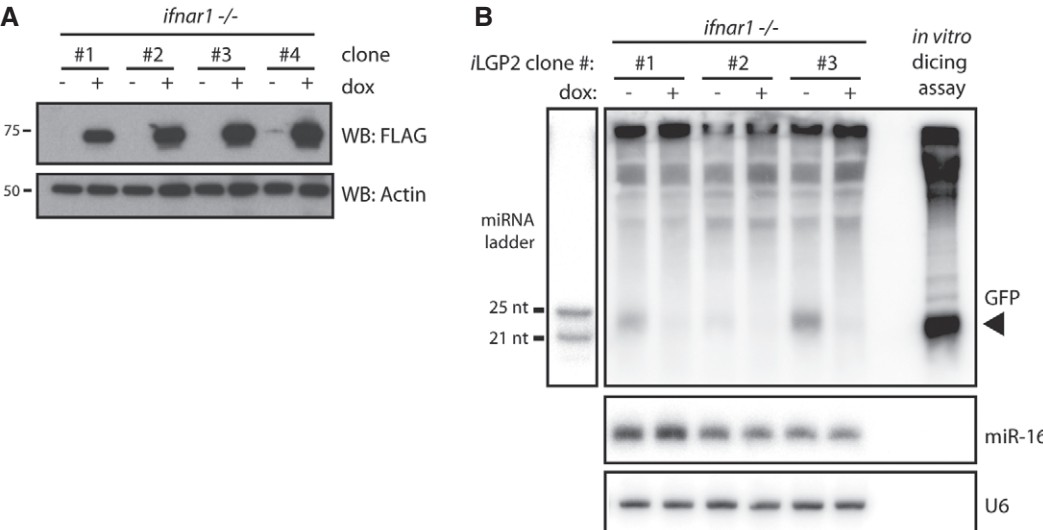

**Figure 5.  LGP2 is sufficient to inhibit dsRNA processing by Dicer in cells lacking a competent type I IFN system.**

A   Immunoblot analysis of four clones of *Ifnar1*$^{-/-}$ MEFs in which expression of FLAG-tagged human LGP2 (iLGP2) is induced following 72 h of doxycycline (dox) treatment. β-Actin serves as loading control.

B   Northern blot analysis of dsRNA-derived siRNAs in *Ifnar1*$^{-/-}$ iLGP2 MEFs left untreated or treated with dox for 24 h prior to transfection with dsRNA-GFP. Twenty-four hours post-transfection, cells were harvested and the generation of siRNAs was analysed by Northern blotting using a probe specific for dsRNA-GFP. The arrow points to dsRNA-GFP-derived siRNAs. RNA from an *in vitro* dicing reaction using dsRNA-GFP was loaded in parallel. A miRNA ladder was used as a size marker, and endogenous U6 served as loading control. The membrane was first probed for dsRNA-GFP, then stripped and reprobed for the miRNA marker, U6 and miR-16.

Source data are available online for this figure.

but not control MEFs following treatment with dsRNA-GFP, although it was less marked and more variable than that observed in cells lacking the IFN receptor (Fig 7). Taken together, these data indicate that LGP2 is an important component of the mechanism by which the IFN response suppresses dsRNAi in differentiated mammalian cells.

# Discussion

Although mammalian somatic cells are fully equipped to process long dsRNA substrates into functional siRNAs, the presence of an intact IFN pathway interferes with such processing explaining why RNAi may not act as major antiviral defence pathway in most mammalian cell types (Maillard *et al*, 2016). Here, we demonstrate that LGP2 represents one of the IFN-inducible factors that inhibits Dicer and blocks the antiviral RNAi response in IFN-competent cells. Our data fit with the idea that the IFN system is incompatible with RNAi in differentiated cells and support the notion that the antiviral capacity of RNAi may be restricted to undifferentiated cell types or niches that have no or little expression of IFN-inducible proteins such as LGP2.

A physical interaction between LGP2 and the Dicer machinery was previously noted in a large-scale study to map the innate immunity interactome (Li *et al*, 2011), although its functional implications were not addressed. Our proteomics approach and subsequent biochemical validation support the earlier findings and indicate that the interaction does not extend to other members of the RLR family. Notably, LGP2 has the highest affinity for dsRNA of the three RLRs (Uchikawa *et al*, 2016). However, our findings indicate that the association with Dicer and inhibition of siRNA generation is not

merely attributable to competition for a common dsRNA substrate. Indeed, the CTD of LGP2, which binds dsRNA and Dicer most strongly, was unable to inhibit Dicer activity. However, this is not to say that dsRNA binding does not contribute to the interaction. LGP2, Dicer, PACT and PKR all contain one or more dsRNA-binding domains, and binding to ligand induces conformational rearrangements that may alter their ability to associate with other proteins (Cole, 2007; Taylor *et al*, 2013; Heyam *et al*, 2015; Uchikawa *et al*, 2016). Our experimental results with different LGP2 domains likely reflect a scenario in which RNA-bound CTD undergoes structural rearrangements that allow it to associate with Dicer, an interaction that is then further strengthened in a cooperative manner by the presence of the helicase domain. This cooperativity is manifest in the fact that only in the context of the entire protein is LGP2 able to inhibit Dicer activity. In contrast to dsRNA processing, we did not find evidence for an inhibitory role for LGP2 on Dicer-dependent pre-miRNA processing, although it cannot be excluded that LGP2 impacts on the processing of a specific subset of pre-miRNAs.

Despite the preservation of the RNAi pathway in mammalian somatic cells, its role as a primary antiviral defence mechanism appears to have been lost during evolution and largely replaced by the protein-based IFN response, with the possible exception of undifferentiated or stem cells. Why is the RNAi antiviral response, so effective in other organisms, attenuated in mammalian somatic cells? The relationship between the RNAi and IFN machinery is only beginning to be defined. A reciprocal inhibition between the two pathways was previously noted in a study that described inactivation of the RISC machinery by poly-ADP-ribosylation of its components following poly(I:C) stimulation or viral infection (Seo *et al*, 2013). It was found that RISC-dependent miRNA processing

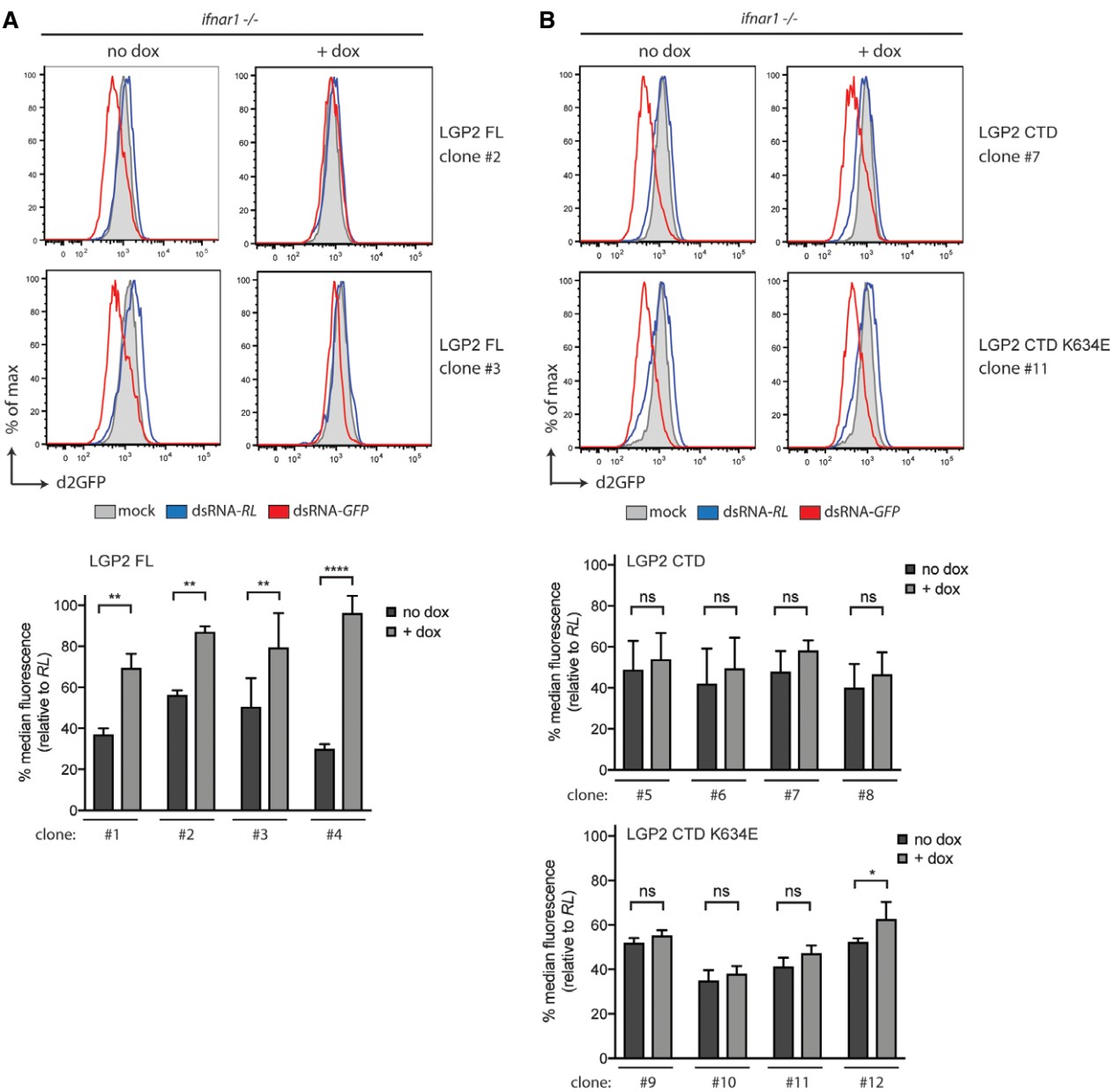

**Figure 6. Expression of full-length LGP2 inhibits dsRNA-mediated RNAi in *Ifnar1*$^{-/-}$ cells.**

A *Ifnar1*$^{-/-}$ iLGP2 cells, which also express a destabilised form of GFP (d2GFP), were transfected with Cy5-labelled long dsRNA corresponding to the first 200 nt of Renilla luciferase (dsRNA-RL) or GFP (dsRNA-GFP) in the absence or presence or doxycycline. Forty-eight hours post-transfection, cells were harvested and d2GFP expression in live, single, Cy5$^+$ cells was analysed by flow cytometry. Histogram plots of two representative clones are shown. Bar graphs display the percentage of GFP median fluorescence intensity of dsRNA-GFP-transfected cells relative to dsRNA-RL-transfected cells in four independent clones.

B Unlike full-length LGP2, LGP2 CTD or CTD K634E expression is unable to suppress dsRNAi in *Ifnar1*$^{-/-}$ MEFs. Four independent clones of *Ifnar1*$^{-/-}$ d2GFP MEFs that inducibly express LGP2 CTD or LGP2 K634E were treated with dsRNA-RL or dsRNA-GFP as in (A). Histogram plot of one representative clone for each construct is shown. Bar graphs display the percentage of GFP median fluorescence intensity of dsRNA-GFP-transfected cells relative to dsRNA-RL-transfected cells in four independent clones.

Data information: All histogram plots are representative of three independent experiments. Each histogram represents a sample size of 10,000 cells. Bar graphs depict the median fluorescence values normalised to those in Renilla-transfected samples. Mean values and SD of three independent experiments are shown. Statistical analysis was performed using two-way ANOVA with Sidak's multiple comparisons test as post-test for pairwise comparisons. Significant differences with Sidak's multiple comparisons test are shown (ns, not significant; *$P$ < 0.05; **$P$ < 0.01; ****$P$ < 0.0001).

represses many ISGs with miRNA-binding sites in their 3′UTRs under normal conditions, whilst inhibition of the RISC machinery through poly-ADP-ribosylation relieved the suppression, allowing

greater ISG protein expression and antiviral activity (Seo *et al*, 2013). Inhibition of the RISC therefore serves to increase the IFN-dependent antiviral response. Likewise, the LGP2-dependent

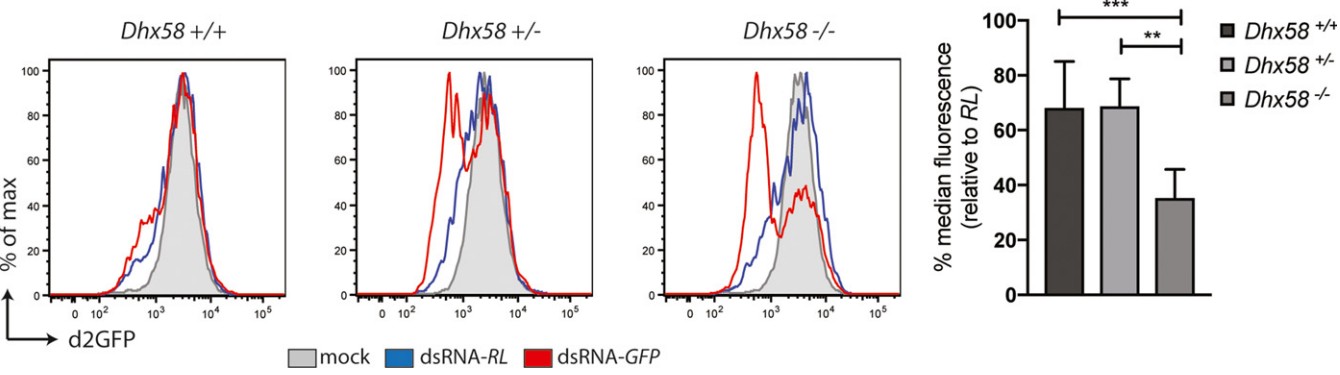

**Figure 7. Genetic ablation of LGP2 (encoded by the gene *Dhx58*) reveals dsRNAi in mammalian cells.**

$Dhx58^{+/+}$, $Dhx58^{+/-}$ and $Dhx58^{-/-}$ MEFs were transduced to stably express the d2GFP reporter. Cells were subsequently transfected with Cy5-labelled dsRNA-RL or dsRNA-GFP and harvested 48 h post-transfection to measure d2GFP expression in live, single, Cy5$^+$ cells by flow cytometry. Histogram plots of one representative experiment are shown and are representative of four independent experiments with duplicate samples. Each histogram represents a sample size of 10,000 cells. Bar graphs display the percentage of GFP median fluorescence intensity of dsRNA-GFP-transfected cells relative to dsRNA-RL-transfected cells. The median fluorescence values were normalised to those in Renilla-transfected samples. Mean values and SD of four independent experiments with duplicate samples are shown. Statistical analysis was performed using one-way ANOVA with Sidaks multiple comparisons test as post-test for pairwise comparisons. Significant differences with Sidaks multiple comparisons test are shown (**$P < 0.01$; ***$P < 0.001$).

inhibition of Dicer processing may protect dsRNA substrates from destruction and preserve them for efficient activation of dsRNA sensors such as MDA5 or ISG proteins such as the dsRNA-binding antiviral proteins PKR and 2′5′-oligoadenylate synthetase. Indeed, the minimal length of dsRNA needed for activation of PKR is 30 nt, suggesting that the ~21-nt product of Dicer processing would interfere with efficient PKR activation (Husain *et al*, 2012). Similarly, a recent report showed that MDA5 and RIG-I, in addition to their canonical role in signalling, function to displace viral proteins pre-bound to dsRNA, thereby freeing the latter for PKR stimulation (Yao *et al*, 2015). Alternatively, rather than being inhibited, the activities of certain components of the RNAi machinery may be redirected to alternate functions in the regulation of IFN induction during viral infection. The dsRNA-binding protein PACT, for example, is important for optimal miRNA processing by Dicer, but also binds the CTD of RIG-I and potentiates activation of IFN production through RIG-I in the context of Sendai virus infection (Kok *et al*, 2011). A similar observation was made for TRBP, which was found to enhance signalling via MDA5 but not RIG-I (Komuro *et al*, 2016), although we failed to substantiate this in our assays. Taken together, these findings point to a scenario in which the activity of the RNAi machinery in mammalian somatic cells is inhibited or redirected in a manner that does not hinder the superior antiviral IFN response.

Curiously, although the IFN pathway is absent in invertebrates, *Caenorhabditis elegans* expresses three Dicer-related helicases (DRH-1, DRH-2 and DRH-3) that bear homology to the helicase domain and the CTD of the mammalian RLRs (Paro *et al*, 2015). DRH-1 and DRH-3 additionally contain a worm-specific N-terminal domain, whilst DRH-2 lacks this domain and is solely composed of the helicase domain and the CTD, thereby exhibiting a remarkable resemblance with LGP2 (Paro *et al*, 2015). Whilst DRH-1 is essential in antiviral, but not exogenous, RNAi in worms, DRH-2 mutants show enhanced RNAi upon exposure to either exogenous or viral dsRNA (Lu *et al*, 2009; Ashe *et al*, 2013; Guo *et al*, 2013). This suggests that DRH-2, through an unknown mechanism, functions as a negative regulator that inhibits the RNAi response in nematodes,

resembling the observations made for its mammalian counterpart LGP2. Our data indicate that, analogous to the worm experiments, genetic ablation of LGP2 reveals dsRNAi in MEFs, although less efficiently and with more variability compared to loss of the IFN response altogether. Further experimentation is therefore needed to address whether LGP2 acts in concert with other ISGs to suppress mammalian RNAi. The recent discovery that human enterovirus 71 encodes a VSR that interferes with Dicer activity to evade RNAi-dependent restriction mechanisms (Qiu *et al*, 2017) opens up future avenues of research to assess the impact of LGP2 on mammalian antiviral RNAi.

# Materials and Methods

### Cell culture and reagents

HEK293 and HeLa cells were cultured in Dulbecco's modified Eagle's medium (DMEM; Gibco, Thermo Fisher Scientific) supplemented with 10% heat-inactivated foetal calf serum (Autogen Bioclear UK, Ltd), 2 mM glutamine and 100 U/ml penicillin/streptomycin (Gibco, Thermo Fisher Scientific) at 10% CO$_2$ at 37°C. $Ifnar1^{-/-}$ MEFs stably expressing d2GFP were prepared previously (Maillard *et al*, 2016) and cultured as described above. F9 embryonal carcinoma cells were obtained from ATCC and cultured as above on tissue culture plates coated with 0.1% gelatin. $Dhx58^{-/-}$ mice were kindly provided by Dr. Michael Gale Jr (Suthar *et al*, 2012) and maintained and bred in accordance with national and institutional guidelines for animal care. $Dhx58^{+/+}$, $Dhx58^{+/-}$ and $Dhx58^{-/-}$ MEFs were generated from day 13.5 embryos from $Dhx58^{+/-}$ intercrosses and immortalised using a minimal dose of a retroviral vector encoding large T antigen as previously described (Schulz *et al*, 2010). All cells were mycoplasma negative. Universal type I IFN alpha (PBL Assay Science) was used at 200 U/ml. Poly(I:C) (Amersham) was used at 1 μg/ml either uncomplexed or complexed with Lipofectamine 2000 (Invitrogen, Thermo Fisher Scientific) at a 1:3 ratio according to manufacturer's instructions. EMCV stocks

were generated as described previously (Deddouche *et al*, 2014), and cells were infected at a multiplicity of infection (MOI) of 1 or 5 in serum-free medium for 1 h, followed by incubation for 5 h (Fig 1D) or overnight (Fig 3B) in complete medium.

### Plasmids, siRNA and transfection

3FLAG-tagged human LGP2, RIG-I and MDA5 were cloned into pcDNA3.1 and verified by Sanger sequencing. 3FLAG-LGP2 NTD #1 (1–486) and NTD #2 (1–552) were generated by introducing three stop codons followed by an additional nucleotide at the appropriate position in the full-length construct using QuikChange mutagenesis (Stratagene). 3FLAG-LGP2 CTD (489–678) was generated by PCR amplification and cloning into pcDNA3.1. The K634E mutation was introduced using QuikChange mutagenesis. The same strategies were used to introduce LGP2, its derivatives, MDA5 or RIG-I in pLVX-tight-puro (Clontech) to allow doxycycline-inducible expression. pIRESneo-FLAG/HA-Ago2 corrected was a gift from Thomas Tuschl [Addgene, #10822 (Meister *et al*, 2004)] and subjected to QuikChange mutagenesis to replace the FLAG epitope with an HA epitope. All plasmid transfections were carried our using Lipofectamine 2000 (Invitrogen, Thermo Fisher Scientific) using manufacturer's instructions. Individual siRNAs or siGenome SmartPools (Dharmacon) were transfected using Dharmafect 1 (Dharmacon) according to the manufacturer's instructions.

### Immunoprecipitation, immunoblotting and antibodies

Cells were washed once in cold PBS and lysed in plates using an NP-40-based lysis buffer [0.5% NP-40, 150 mM NaCl, 50 mM Tris pH 7.5, 5 mM MgCl$_2$, 10% glycerol, protease inhibitors (Complete EDTA-free, Roche) and phosphatase inhibitor cocktail set V (Calbiochem)] for 30 min at 4°C. Cell lysates were cleared by centrifugation (14,500 × *g*, 15 min, 4°C), and protein content was determined by BCA Protein Assay (Pierce, Thermo Fisher Scientific). Lysates were subjected to FLAG immunoprecipitation using FLAG M2 agarose beads or FLAG M2 magnetic beads (both Sigma) for 4 h or overnight at 4°C in the presence or absence of 200 U/ml RNase I, 10 μg/ml RNase A or 5 U/ml RNase III (all from Ambion). Where required 10% of the lysates (pre- and post-immunoprecipitation) were subjected to RNA extraction using TRIzol LS (Invitrogen, Thermo Fisher Scientific) and analysis on an Agilent 2100 Bioanalyzer to monitor efficient RNase activity. Immunoprecipitates were washed four times in lysis buffer and resolved on 4–20% miniprotean TGX precast gels (Bio-Rad), transferred to Immobilon-P PVDF membrane (Millipore) by semi-dry transfer, blocked in 5% non-fat dried milk in PBS-T (PBS, 0.1% Tween-20) and incubated with the relevant primary and secondary antibodies: Dicer (D38E7), PKR (#3072S), MDA5 (D74E4), β-actin-HRP (13E5; all from Cell Signalling Technology), PACT (EPR3224, Novus Biologicals), TRBP2 (D-5, Santa Cruz), HA-HRP (clone 3F10, Roche), FLAG-HRP (clone M2, Sigma), capsid protein VP1 from EMCV (purchased from Dr. Emiliana Brocchi's laboratory, Brescia, Italy), goat anti-mouse IgG-HRP (Molecular Probes, Invitrogen) and goat anti-rabbit IgG-HRP (Southern Biotech). For detection, the following substrates were used: SuperSignal West Pico Chemiluminescent Substrate (Thermo Fisher Scientific) and Luminata Crescendo or Luminata Forte Western HRP substrate (both Millipore). For RNA

immunoprecipitations, RNA was extracted from immunoprecipitates using TRIzol or TRIzol LS (Invitrogen, Thermo Fisher Scientific).

### LC-MS/MS analysis

Proteins were eluted from the beads using 0.5 mg/ml 3xFLAG peptide (Sigma), concentrated on a Vivaspin 500 with a 5-kDa cut-off (Sartorius) and subjected to SDS–PAGE. Next, proteins were overnight in-gel digested using a Janus liquid handling system (Perkin Elmer). Peptide mixtures were analysed using an LTQ-Orbitrap XL mass spectrometer (Thermo Fisher Scientific) coupled to a nano-acquity UPLC (Waters Corporation). Raw data processing was performed using MaxQuant v1.3.05. Intensity-based absolute quantification (iBAQ) was used for protein quantification. Data were visualised in Perseus. The mass spectrometry proteomics data have been deposited to the ProteomeXchange Consortium via the PRIDE partner repository with the dataset identifier PXD008364 (Vizcaíno *et al*, 2016).

### Luciferase reporter assay

$1.25 \times 10^5$ HEK293 cells/well were plated in 24-well plates, transfected with 125 ng of p125-Luc, encoding a Firefly luciferase reporter gene under the control of the IFN-β promotor (gift from T. Fujita, Kyoto University, Japan) and 50 ng pRL-TK (Promega) using Lipofectamine 2000, and treated with 200 U/ml universal type I IFN alpha (PBL Assay Science). Six-eight hours later, cells were transfected with water (mock) or with the indicated amounts of total RNA extracted from HeLa cells that were infected with EMCV overnight (Fig 3A) or with RNA extracted from LGP2 immunoprecipitates (Fig 3B). Luciferase activity was measured 16–17 h later using the dual-luciferase assay reporter system (Promega). Firefly luciferase values were divided by Renilla luciferase values to normalise for transfection efficiency. All data are shown as fold increase relative to mock-transfected cells.

### Preparation of long dsRNA

Long dsRNA of 200 nt corresponding to the GFP or Renilla coding sequence was synthesised by *in vitro* transcription with T7 RNA polymerase (T7 MEGAscript kit, Ambion) as detailed previously (Maillard *et al*, 2016). RNA was labelled by incorporation of 1/10$^{th}$ of Cy5-CTP (Amersham CyDye Fluorescent Nucleotides Cy5-CTP, GE Healthcare Life Sciences) in the IVT reaction. Following Turbo DNAse treatment (Invitrogen, Thermo Fisher Scientific), removal of unincorporated nucleotides using an Illustra G-25 column (GE Healthcare) and phenol/chloroform extraction, the RNA was annealed and single-stranded RNA molecules were removed by RNase I treatment.

### Protein purification

Full-length 3FLAG-tagged human LGP2, RIG-I, MDA5 and LGP2 CTD (489–678) were cloned into pBacPAK-His3-GST. The K634E mutation was introduced by QuikChange mutagenesis (Stratagene). All constructs were verified by Sanger sequencing. The plasmids were transfected into insect cells (Sf21 Berger) using the FlashBac system (Oxford Expression Technologies) and Fugene HD (Promega)

according to manufacturer's instructions. The cells were incubated for 5 days at 27°C. The resulting 2 ml P1 virus was used to infect 25 ml ($1 \times 10^6$ cells/ml) of Sf21 cells for 3 days at 27°C. The resulting P2 virus was titrated using qPCR and amplified to 50 ml ($1 \times 10^6$ cells/ml, MOI 1) of P3. The P3 virus was used to infect a large scale of Sf21 cells ($1 \times 10^6$ cells/ml, MOI 1). Cells were lysed in 50 mM Tris pH 7.5, 150 mM NaCl, 5% glycerol, 1 mM DTT, 0.5% Triton X-100, Benzonase (Sigma) and protease inhibitors (Roche) by sonication on ice. The lysate was cleared by centrifugation (20,000 $g$ for 10 min at 4°C) and incubated with Glutathione Agarose (Cube Biotech) at 4°C for 2 h. Using a Superdex 75 10/30 GL column (GE Healthcare), the resin was washed three times with 10 column volumes of 50 mM Tris pH 7.5, 150 mM NaCl, 5% glycerol and 1 mM DTT. The proteins were cleaved of the resin using 3C protease at 4°C overnight and further purified using gel filtration in 50 mM Tris pH 7.5, 150 mM NaCl, 5% glycerol. The fractions containing pure LGP2 were pooled and concentrated.

### In vitro dicing assay

Human FLAG-Dicer in pCAGGS [a gift from Phil Sharp, Addgene #41584 (Gurtan *et al*, 2012)] was transfected into HEK293T cells using Lipofectamine 2000. Twenty-eight hours post-transfection, cells were lysed in lysis buffer [30 mM Tris pH 6.8, 50 mM NaCl, 3 mM $MgCl_2$, 5% glycerol, 0.4% NP-40 and protease inhibitors (Roche)], and the lysate was cleared by centrifugation. FLAG-Dicer was retrieved using FLAG M2 beads (Sigma), and the immunoprecipitates were washed 5× in lysis buffer and once in lysis buffer without NP-40 followed by elution using 0.5 mg/ml 3xFLAG peptide (Sigma) in lysis buffer without NP-40. The purity and concentration of the eluates were assessed by SDS–PAGE and Coomassie staining using BSA as a standard. For dicing assays, FLAG-Dicer was incubated with 50 nM dsRNA or 100 nM synthetic Cy5-labelled pre-Let7a-1 (custom made at Dharmacon, incorporating Cy5-A at position 10 and 59) and FLAG-tagged LGP2, MDA5, RIG-I, CTD or CTD K634E in dicing buffer [30 mM Tris pH 6.8, 50 mM NaCl, 3 mM $MgCl_2$, 5% glycerol, 1 mM DTT, RNAsin (Promega)] for 1 h at 37°C followed by phenol/chloroform purification. The RNA was resuspended in formamide sample buffer without xylene blue (47.5% formamide, 0.01% SDS, 0.01% bromophenol blue, 0.5 mM EDTA), loaded onto a 15% Novex TBE-Urea gel (Thermo Fisher Scientific) and visualised by in-gel fluorescence on an ImageQuant LAS 4000 (GE Healthcare).

### Lentiviral production and transduction

To generate d2GFP-expressing cell lines, cells were transduced with lentiviral vectors encoding d2GFP (pRRLsin PGK d2GFP) and sorted on a BD FACSAria (Fusion) as described previously (Maillard *et al*, 2016). 3FLAG-hLGP2, CTD and CTD K634E were cloned in pLVX-tight-puro lentiviral plasmid, which is compatible with the Lenti-X Tet-On Advanced Lentiviral expression system (Clontech). Lentiviral particles were produced by transfecting 20 μg of the above plasmids combined with 6 μg of pMD2.G (Addgene, 12259), encoding the vesicular stomatitis virus G protein, and 14 μg of psPAX2 (Addgene, 12260) into 70–80% confluent HEK293T cells in a 15-cm dish using TransIT-293 (Mirus). The pMD2.G and psPAX2 packaging plasmids were a gift from Didier Trono. Medium was changed 16 h post-transfection, and supernatants were collected at 48–72 h post-transfection, passed through a 0.45-μm filter, concentrated by ultracentrifugation for 2 h at 4°C and resuspended in PBS. Cells were transduced in a 24-well format, and as the targeting and immortalisation procedure of the *Ifnar1*$^{-/-}$ MEFs prevented puromycin- or neomycin-based antibiotic selection, cells were subsequently subjected to single-cell cloning. Clones in which the FLAG-tagged construct was induced following treatment for 48 h with 1 μg/ml doxycycline (Sigma) were identified by intracellular flow cytometry and immunoblotting and expanded. F9 cells were selected by neomycin and puromycin following transduction with the Lenti-X Tet-On Advanced Lentiviral system to express LGP2 in an inducible manner.

### Northern blot

Mouse embryonic fibroblasts were plated in 10-cm dishes and treated with or without 1 μg/ml doxycycline (Sigma) overnight followed by transfection with 4.5 μg of dsRNA-GFP using an RNA: Lipofectamine 2000 ratio of 1:6. Twenty-four hours post-transfection, total RNA was extracted using TRIzol (Thermo Fisher Scientific) according to the manufacturer's instructions. 16 μg of RNA was loaded onto a denaturing 17.5% polyacrylamide/urea gel and further processed as described (Maillard *et al*, 2013, 2016).

### Flow cytometry and intracellular staining

For the d2GFP reporter assay, $4 \times 10^4$ cells/well were plated in a 24-well plate and treated with or without 1 μg/ml doxycycline (Sigma). The next day, 150–200 ng of Cy5-labelled dsRNA-RL or dsRNA-GFP was transfected into cells using a dsRNA:Lipofectamine 2000 ratio of 1:6. Forty-eight hours later, cells were trypsinised, washed in PBS and resuspended in FACS buffer (PBS, 2% FCS, 2 mM EDTA) supplemented with the live cell/dead cell discriminator dye DAPI. For intracellular staining, cells were fixed and permeabilised (Fix & Perm kit 1000, Nordic MUbio), stained for 1 h with a Cy3-labelled FLAG M2 antibody (Sigma) and washed 2× in FACS buffer. All flow cytometric analyses were performed with an LSR Fortessa (BD Biosciences) acquiring a sample size of 10,000 life cells to meet statistical robustness. Data were analysed using FlowJo (Tree Star).

### RT–PCR

Total RNA was extracted using RNeasy Mini Kit columns with DNase treatment according to the manufacturer's instructions (Qiagen). Five hundred nanogram of RNA was reverse transcribed using random primers and SuperScript II Reverse Transcriptase (Thermo Fisher Scientific). cDNA was diluted in nuclease-free water, and gene expression was analysed by qPCR using *DHX58* and *ACTB* TaqMan Gene Expression Assays (Thermo Fisher Scientific) and ABI 7500 FAST machines (Applied Biosystems).

### Statistical analysis

Statistical analysis was performed using GraphPad Prism 7. A *P*-value < 0.05 was considered statistically significant.

**Expanded View** for this article is available online.

## Acknowledgements

We thank Dr. Michael Gale Jr for the gift of *Dhx58*$^{-/-}$ mice. We also thank all members of the Immunobiology Laboratory for helpful discussions and suggestions, Oliver Gordon for advice on statistical analysis and the Crick Equipment Park and the Flow Cytometry Facility for technical assistance. This work was supported by The Francis Crick Institute, which receives its core funding from Cancer Research UK (FC001136), the UK Medical Research Council (FC001136) and the Wellcome Trust (FC001136) and by a prize from the Fondation Louis-Jeantet. A.G.V. was supported by an EMBO Long-Term Fellowship and a Rubicon Fellowship from the Netherlands Organization for Scientific Research. P.V.M. was supported by an Advanced Postdoc Mobility fellowship from the Swiss National Science Foundation and by a Marie-Curie Actions Intra-European fellowship.

## Author contributions

AGV and CRS designed experiments, analysed data and wrote the manuscript. AGV conducted experiments with assistance from PVM, JMS and SAL. SD-G provided key reagents. AB and SK generated insect-purified recombinant proteins. APS assisted with the LC-MS/MS analysis. CRS supervised the project.

## Conflict of interest

The authors declare that they have no conflict of interest.

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
