## [Review Process File · The EMBO Journal]

The RIG-I like receptor LGP2 inhibits Dicer-dependent processing of long double-stranded RNA and blocks RNA interference in mammalian cells.

Annemarte G. van der Veen, Pierre V. Maillard, Jan Marten Schmidt, Sonia A. Lee, Safia Deddouche-Grass, Annabel Borg, Svend Kjær, Ambrosius P. Snijders, Caetano Reis e Sousa

Review timeline:

Submission date:	1 June 2017
Editorial Decision:	13 July 2017
Revision received:	2 November 2017
Editorial Decision:	30 November 2017
Revision received:	8 December 2017
Accepted:	15 December 2017

Editor: Karin dumstrei

Transaction Report:

1st Editorial Decision

13 July 2017

Thanks for submitting your manuscript to the EMBO Journal. Your study has now been seen by three referees and their comments are provided below.

As you can see from the comments, the referees find the analysis interesting. However, they also find that further revisions are needed to further substantiate the conclusions. The referees' comments are very constructive and they bring up a number of points that I anticipate you will be able to address. In particular the LGP2 loss of function analysis is an important point to resolve.

Given the referees' comments I would like to invite you to submit a revised version of the manuscript, addressing the comments of all three reviewers. Let me know if we need to discuss any points in further detail.

When preparing your letter of response to the referees' comments, please bear in mind that this will form part of the Review Process File, and will therefore be available online to the community. For more details on our Transparent Editorial Process, please visit our website: http://emboj.embopress.org/about#Transparent_Process

Thank you for the opportunity to consider your work for publication. I look forward to your revision.

REFeree REPORTS

Referee #1:

In this manuscript, Reis e Sousa and colleagues report that the RIG-I like receptor LGP2 interacts with Dicer in mammalian cells and inhibits long dsRNA processing and RNA interference (RNAi) triggered by dsRNA, without affecting processing of pre-miRNAs. The important biological question lying behind these investigations is that of the apparent incompatibility between dsRNA triggered RNAi and the interferon (IFN) system in most mammalian cells. The paper is well written, the experiments include the appropriate controls, the data are clear and well presented. I have two concerns concerning the interpretation made by the authors: the first is that the conclusions drawn are only based on experiments involving overexpression of LGP2. If LGP2 is indeed a relevant inhibitor of Dicer activity, one would expect an improvement of dsRNA processing by Dicer in LGP2 knock-out cells. In case of a negative result, this would not necessarily invalidate the theory of the authors, but it would mean that other factors contribute and that a note of caution (in particular in the title) is required. The second concern is that no mechanism is provided, which makes the manuscript essentially descriptive.

Specific comments:

- 1) LGP2 interacts with Dicer via CTD : the FLAG-CTD-K634E reduces, but does not abrogate, I suggest revision of wording
- 2) Dicer is not required for IFN signalling. The results are clear, but a statistical analysis would be important . The results with TRBP are not necessarily inconsistent since the current experiments are for a depletion of TRBP while in the other paper (Komuro 2016) the effects were seen with overexpression of the protein.
- 3) LGP2 inhibits Dicer : some statistics are required for the claims re. fig 4d
- 4) LGP2 is sufficient to inhibit processing of dsRNA's by Dicer in *Ifnar* $-/-$ cells : the claim is supported by data, but it is not shown that this effect is specific to LGP2 (controls with MDA5/RIG-I OE should not abolish this processing). Clone II has limited production of 21nt RNAs.
- 5) LGP2 is sufficient to block dsRNAi in *Ifnar* $-/-$ cells : same comment as above, what is the effect of OE of MDA5 and RIG-I ? No statistics. There seems to be a small effect of the dox treatment, i.e., a suppression of dicer activity, in the CTD experiments for some of the clones (#7, #8, #12). Was this formally tested?
- 6) Some unexplained abbreviations (PKR, PACT, TRBP)

Referee #2:

In this manuscript, van der Veen et al. provide a novel explanation as to why anti-viral RNAi (dsRNAi) is inhibited in mammalian cells. This largely represents a follow-up to previous work in the EMBO Journal from the same group, which established that the IFN response masks dsRNAi activity in a manner that precludes the dsRNAi machinery from exerting antiviral effector functions (Maillard et al, 2016). In the current work, the authors extend these finding to focus on a specific interferon stimulated gene (ISG), known as LGP2, which appears to inhibit dsRNAi. Specifically, the authors find that LGP2 engages Dicer, a core component of the RNAi machinery, to inhibit Dicer processing of siRNAs. Overall, this is a well-executed manuscript of general interest to the small RNA / innate immunity / and virus communities, and is well suited for the Journal. Still there are some issues that warrant further clarification. Below are comments and suggestions to further improve this manuscript.

Major comments:

Does IFN indeed induce LGP2 in the contexts tested? Given that LGP2 interacts with many RNAi components at steady-state (Fig 1F), it is unclear how interferon stimulation augments Dicer function specifically via LGP2. That the authors observe a specific effect for LGP2 in inhibiting siRNA processing, but not miRNA processing, is quite interesting, as implies functionally distinct roles for LGP2 depending on (presumably) binding to a dsRNA substrate. Has this been directly tested? The authors' discussion point that LGP2 dsRNA recognition may shield the dsRNA from being a Dicer substrate is appropriate, and should be tested. Testing LGP2 mutants that cannot bind dsRNA, but that can interact with Dicer, may assist in these studies.

In addition, I highly recommend that the authors verify that miRNA processing is not affected by LGP2 activity in cells. The experiments in Figure 5, modified to observe miRNA processing, would suffice. This would strengthen the arguments made based on the in vitro dicing experiments.

Are LGP2 knockdown / knockout / depleted cells deficient in inhibiting Dicer? In other words, is LGP2 alone sufficient for this effect? Relatedly, can lack of LGP2 unmask dsRNAi in IFN signaling capable cells? I realize that this is the cliffhanger the manuscript concludes on, but a bit more discussion is needed beyond being "unclear at present."

Lastly, while the flow cytometry data is convincing, it would be ideal to show that inhibiting dsRNAi by LGP2 impacts viral RNA levels directly. In other words, in a biologically relevant setting, is an unmasked anti-viral RNAi response suppressed by LGP2?

Minor comments:

For visual clarity, axis font sizes and styles (boldness, etc) should be made consistent across all graphs.

Referee #3:

In their study "The RIG-I like receptor LGP2 inhibits Dicer-dependent processing of long double-stranded RNA and blocks RNA interference in mammalian cell", van der Veen, et al. present evidence that the RLR LGP2 interacts with Dicer and several known Dicer cofactors and that overexpression of LGP2 inhibits Dicer activity. Given the evidence to date for antagonism between the type-I IFN system and RNAi, identifying candidate ISGs that mediate this phenomenon is of great interest. However, this reviewer is not convinced that enough data has been presented to fully substantiate the authors' claims and would like to see several major points addressed (see below). In the past, data on LGP2 obtained via overexpression has contradicted what was later found in LGP2-deficient mice (cf. (Rothenfusser et al., 2005) and (Satoh et al., 2010)). Thus, it would be of particular importance to investigate LGP2-deficient cell lines (e.g. CRISPR/Cas9 or MEFs from the LGP2^{-/-} mouse) to see if RNAi is restored or enhanced and provide genetic proof for the mechanism the authors propose.

Major points

Figure 5/6:

- The authors mention that constitutive overexpression of LGP2 was "detrimental" to the cells. If that is the case, then the toxicity after 48h of strong overexpression should also be investigated. Cytotoxicity could non-specifically affect the cells ability to perform a number tasks including Dicer-induced dsRNA processing.
- Overexpression of the LGP2 CTD should also be included in Figure 5 as a control for this experiment. Moreover, cytotoxicity data for LGP2 CTD overexpression should be included.
- These assays should be extended to include other cell lines that have been reported to process long dsRNA for RNAi, e.g. P19 (ATCC® CRL-1825 {trade mark, serif}) or F9 (ATCC® CRL-1720 {trade mark, serif}) from (Billy et al., 2001). This would perhaps solve the problem with cytotoxicity in MEFs and demonstrate that the relevance goes beyond one cell line.

LGP2-deficient cell lines:

- In order to assess the relevance of endogenous LGP2 expression for RNAi, evaluating LGP2-deficient cells would be of the utmost importance.
 - o Is it possible to restore RNAi in differentiated cells if LGP2 is knocked out?
 - o In MEFs do LGP2^{-/-} show the same RNAi activity as IFNAR^{-/-}? Does the additional KO of LGP2 via CRISPR/Cas9 enhance what is already observed in IFNAR^{-/-} MEFs?
 - o Is the existing RNAi in cell lines such as P19 or F9 enhanced by LGP2 deficiency?

Minor Points

Figure 5/6:

- IFNAR^{-/-}-MEFs without LGP2 but with doxycycline treatment should be included in both figures (or the supplementary figures).

LGP2 Expression:

- Has the correlation between basal LGP2 levels and RNAi activity been investigated? How much LGP2 is expressed in IFNAR^{-/-} MEFs?

References

- Billy, E., Brondani, V., Zhang, H., Müller, U., and Filipowicz, W. (2001). Specific interference with gene expression induced by long, double-stranded RNA in mouse embryonal teratocarcinoma cell lines. *Pnas* 98, 14428-14433.
- Rothenfusser, S., Goutagny, N., DiPerna, G., Gong, M., Monks, B.G., Schoenemeyer, A., Yamamoto, M., Akira, S., and Fitzgerald, K.A. (2005). The RNA helicase Lgp2 inhibits TLR-independent sensing of viral replication by retinoic acid-inducible gene-I. *The Journal of Immunology* 175, 5260-5268.
- Satoh, T., Kato, H., Kumagai, Y., Yoneyama, M., Sato, S., Matsushita, K., Tsujimura, T., Fujita, T., Akira, S., and Takeuchi, O. (2010). LGP2 is a positive regulator of RIG-I- and MDA5-mediated antiviral responses. *Proc. Natl. Acad. Sci. U.S.A.* 107, 1512-1517.

1st Revision - authors' response

2 November 2017

Referee #1:

In this manuscript, Reis e Sousa and colleagues report that the RIG-I like receptor LGP2 interacts with Dicer in mammalian cells and inhibits long dsRNA processing and RNA interference (RNAi) triggered by dsRNA, without affecting processing of pre-miRNAs. The important biological question lying behind these investigations is that of the apparent incompatibility between dsRNA triggered RNAi and the interferon (IFN) system in most mammalian cells. The paper is well written, the experiments include the appropriate controls, the data are clear and well presented. I have two concerns concerning the interpretation made by the authors: the first is that the conclusions drawn are only based on experiments involving overexpression of LGP2. If LGP2 is indeed a relevant inhibitor of Dicer activity, one would expect an improvement of dsRNA processing by Dicer in LGP2 knock-out cells. In case of a negative result, this would not necessarily invalidate the theory of the authors, but it would mean that other factors contribute and that a note of caution (in particular in the title) is required. The second concern is that no mechanism is provided, which makes the manuscript essentially descriptive.

We thank the reviewer for his/her positive and constructive feedback. We are pleased to include new data in this revision on the effect of LGP2 deficiency on dsRNAi. As predicted by the reviewer, we find that loss of LGP2 in MEFs reveals dsRNAi, although with more variability and less strongly than loss of IFNAR. We have included these data as a new Figure 7.

It is important to explain why this result was not included in the first submission. We had previously tested LGP2-deficient MEFs and had not seen recovery of dsRNAi. This was stated in the last sentence of our original discussion as “data not shown” and led us to conclude that “LGP2 is likely to be only one of several ISGs able to suppress mammalian RNAi”. While we believe that that conclusion is still appropriate, it now seems that loss of LGP2 can, in some instances, be sufficient to reveal dsRNAi. Interestingly, we had generated the initial set of LGP2-deficient MEFs from the LGP2-deficient mouse strain described in Satoh et al (PNAS 2010). Worryingly, even though we got a few embryos to develop to E13.5 (and were therefore able to generate MEFs), we failed to obtain any live LGP2 knockout progeny mice. The embryonic lethality was unexpected as the original Satoh et al paper stated that LGP2-deficient mice are viable even if born at lower Mendelian ratios than predicted. It raised concerns regarding the mouse strain we had obtained and the reliability of the data obtained with the initial set of MEFs. We therefore obtained an independently generated LGP2-deficient mouse strain from the lab of Michael Gale (Suthar, Immunity 2012). Intercrosses of this strain of LGP2 heterozygous mice resulted in live knockout progeny at

Mendelian ratios, which gave us confidence to proceed and generate a new set of transformed MEFs from littermate embryos of each genotype (+/+, +/- and -/-). When these new MEFs were tested, we found that the -/- MEFs displayed dsRNAi (as measured by specific GFP downregulation in our assay) although we also noticed that the extent of the phenomenon (i.e., magnitude of downregulation) was less marked than that seen with *Ifnar*^{-/-} MEFs. The data compiled from four independent experiments are now presented in a new Figure 7.

Specific comments:

1) *LGP2 interacts with Dicer via CTD : the FLAG-CTD-K634E reduces, but does not abrogate, I suggest revision of wording*

We have rephrased this sentence as suggested by the reviewer.

2) *Dicer is not required for IFN signalling. The results are clear, but a statistical analysis would be important . The results with TRBP are not necessarily inconsistent since the current experiments are for a depletion of TRBP while in the other paper (Komuro 2016) the effects were seen with overexpression of the protein.*

We have now included the following note regarding statistical analysis in the legend of Figure 3A: “Statistical analysis was performed using one-way ANOVA with Sidak’s multiple comparisons test as post-test for pairwise comparisons using untransfected cells as control for each dose. Significant differences were observed for siMDA5 only (**P* < 0.05).”

We have changed the text in the manuscript to clarify that our observations were made using siRNA-mediated depletion of TRBP while Komuro et al. based their conclusions on TRBP overexpression experiments.

3) *LGP2 inhibits Dicer : some statistics are required for the claims re. fig 4d*

We have now included the appropriate statistics in Fig. 4C to support our claim that LGP2 inhibits Dicer activity, while MDA5 and RIG-I have no or minimal effect, respectively.

4) *LGP2 is sufficient to inhibit processing of dsRNA's by Dicer in Ifnar -/- cells : the claim is supported by data, but it is not shown that this effect is specific to LGP2 (controls with MDA5/RIG-I OE should not abolish this processing). Clone II has limited production of 2Int RNAs.*

5) *LGP2 is sufficient to block dsRNAi in Ifnar -/- cells : same comment as above, what is the effect of OE of MDA5 and RIG-I ? No statistics. There seems to be a small effect of the dox treatment, i.e., a suppression of dicer activity, in the CTD experiments for some of the clones (#7, #8, #12). Was this formally tested?*

We have now generated *Ifnar*^{-/-} cells in which RIG-I and MDA5 can be expressed in a doxycycline-inducible manner. We find that inducible RIG-I or MDA5 overexpression has no effect on dsRNAi in four individual clones tested, as determined in our standard d2GFP-based reporter assay. We have incorporated these data as Supplemental Figure 5.

We have now included statistics in Figure 6.

Finally, the impact of CTD expression on Dicer activity was further tested at the level of small RNA production by Northern blot (new Supplemental Figure 4E). These data confirm that the CTD of LGP2 by itself is unable to markedly suppress dsRNA processing by Dicer. However, as we cannot formally prove the absence of any residual activity, we have rephrased the text accordingly.

6) *Some unexplained abbreviations (PKR, PACT, TRBP)*

We have now included the full names of these proteins.

Referee #2:

In this manuscript, van der Veen et al. provide a novel explanation as to why anti-viral RNAi (dsRNAi) is inhibited in mammalian cells. This largely represents a follow-up to previous work in the EMBO Journal from the same group, which established that the IFN response masks dsRNAi activity in a manner that precludes the dsRNAi machinery from exerting antiviral effector functions (Maillard et al, 2016). In the current work, the authors extend these finding to focus on a specific

interferon stimulated gene (ISG), known as LGP2, which appears to inhibit dsRNAi. Specifically, the authors find that LGP2 engages Dicer, a core component of the RNAi machinery, to inhibit Dicer processing of siRNAs. Overall, this is a well-executed manuscript of general interest to the small RNA / innate immunity / and virus communities, and is well suited for the Journal. Still there are some issues that warrant further clarification. Below are comments and suggestions to further improve this manuscript.

We thank the reviewer for his/her positive and constructive feedback on our manuscript.

Major comments:

Does IFN indeed induce LGP2 in the contexts tested? Given that LGP2 interacts with many RNAi components at steady-state (Fig 1F), it is unclear how interferon stimulation augments Dicer function specifically via LGP2. That the authors observe a specific effect for LGP2 in inhibiting siRNA processing, but not miRNA processing, is quite interesting, as implies functionally distinct roles for LGP2 depending on (presumably) binding to a dsRNA substrate. Has this been directly tested? The authors' discussion point that LGP2 dsRNA recognition may shield the dsRNA from being a Dicer substrate is appropriate, and should be tested. Testing LGP2 mutants that cannot bind dsRNA, but that can interact with Dicer, may assist in these studies.

While we agree with the reviewer that it would be very interesting to further dissect the mechanism by which LGP2 inhibits dsRNA processing by Dicer, we have been unable to find LGP2 mutants that bind Dicer but not dsRNA. As such, at present, we have no obvious means of testing the shielding model.

In addition, I highly recommend that the authors verify that miRNA processing is not affected by LGP2 activity in cells. The experiments in Figure 5, modified to observe miRNA processing, would suffice. This would strengthen the arguments made based on the in vitro dicing experiments.

We have now tested the impact of LGP2 on miRNA processing, as suggested by the reviewer. We examined miR-16 expression by Northern Blot (Figure 5) and find that LGP2 has no impact on its levels. These data complement those obtained in vitro with let7a. We can, however, not exclude the possibility that LGP2 may impact the processing of a subset of miRNAs by Dicer. We have noted this briefly in the discussion.

Are LGP2 knockdown / knockout / depleted cells deficient in inhibiting Dicer? In other words, is LGP2 alone sufficient for this effect? Relatedly, can lack of LGP2 unmask dsRNAi in IFN signaling capable cells? I realize that this is the cliffhanger the manuscript concludes on, but a bit more discussion is needed beyond being "unclear at present."

We are pleased to be able to include new data on the effect of LGP2 deficiency on dsRNAi in the revised version of the manuscript. We kindly refer the reviewer to the response to point 1 of Reviewer 1.

Lastly, while the flow cytometry data is convincing, it would be ideal to show that inhibiting dsRNAi by LGP2 impacts viral RNA levels directly. In other words, in a biologically relevant setting, is an unmasked anti-viral RNAi response suppressed by LGP2?

We fully agree with the reviewer that it would be desirable to test the impact of LGP2 on viral RNA levels directly. Such an experiment has been difficult due to the unavailability of a mammalian virus that lacks a viral suppressor of RNAi (VSR) and is unambiguously restricted by anti-viral RNAi. To circumvent this, we have previously employed a recombinant Semliki Forest Virus engineered to encode the Renilla luciferase coding sequence in its genome. Treatment of *Ifnar*^{-/-} cells with dsRNA encoding Renilla prior to SFV infection was shown to inhibit viral replication (Maillard et al, EMBO J 2016). However, this approach to demonstrate anti-viral RNAi is indirect and lacks physiological relevance. The recent discovery of a mammalian virus (HEV71) that encodes a VSR that blocks RNAi-mediated restriction of viral replication (Qiu et al, Immunity, June 2017) opens up new possibilities to study how the IFN pathway and its components affect the propagation of wild type HEV71 or its VSR-deficient derivative. We intend to explore how LGP2 impacts on HEV71 replication in future studies and mention this briefly at the end of the revised Discussion.

Minor comments:

For visual clarity, axis font sizes and styles (boldness, etc) should be made consistent across all graphs.

We have now altered all graphs to make the layout more consistent.

Referee #3:

In their study "The RIG-I like receptor LGP2 inhibits Dicer-dependent processing of long double-stranded RNA and blocks RNA interference in mammalian cell", van der Veen, et al. present evidence that the RLR LGP2 interacts with Dicer and several known Dicer cofactors and that overexpression of LGP2 inhibits Dicer activity. Given the evidence to date for antagonism between the type-I IFN system and RNAi, identifying candidate ISGs that mediate this phenomenon is of great interest. However, this reviewer is not convinced that enough data has been presented to fully substantiate the authors' claims and would like to see several major points addressed (see below). In the past, data on LGP2 obtained via overexpression has contradicted what was later found in LGP2-deficient mice (cf. (Rothenfusser et al., 2005) and (Sato et al., 2010)). Thus, it would be of particular importance to investigate LGP2-deficient cell lines (e.g. CRISPR/Cas9 or MEFs from the LGP2^{-/-} mouse) to see if RNAi is restored or enhanced and provide genetic proof for the mechanism the authors propose.

We thank the reviewer for his/her constructive feedback on our manuscript. We are pleased to be able to include new data on the effect of LGP2 deficiency on dsRNAi in the revised version of the manuscript. We kindly refer the reviewer to the response to point 1 of Reviewer 1.

Major points

Figure 5/6:

- The authors mention that constitutive overexpression of LGP2 was "detrimental" to the cells. If that is the case, then the toxicity after 48h of strong overexpression should also be investigated. Cytotoxicity could non-specifically affect the cells ability to perform a number tasks including Dicer-induced dsRNA processing.

We apologise for the our mis-statement and the confusion it has caused. What we observed is that constitutive LGP2 expression decreases over time in the transduced cell population, which indicates that the transduced cells have a disadvantage compared to cells with low or no LGP2 expression. The inducible expression of LGP2 does not have any significant impact on cell viability after 72 h of doxycycline treatment, as illustrated in the graph below. We have rephrased the text in the manuscript for clarification.

- *Overexpression of the LGP2 CTD should also be included in Figure 5 as a control for this experiment. Moreover, cytotoxicity data for LGP2 CTD overexpression should be included.*
 We have now investigated by Northern Blot the impact of the LGP2 CTD on small RNA production and incorporated this experiment as a new Supplemental Figure 4C. The data confirm that the CTD of LGP2 does not suppress dsRNA processing by Dicer. LGP2 CTD overexpression is not cytotoxic (see reply to previous comment).

- *These assays should be extended to include other cell lines that have been reported to process long dsRNA for RNAi, e.g. P19 (ATCC® CRL-1825{trade mark, serif}) or F9 (ATCC® CRL-1720{trade mark, serif}) from (Billy et al., 2001). This would perhaps solve the problem with cytotoxicity in MEFs and demonstrate that the relevance goes beyond one cell line.*

We have now addressed the effect of inducible LGP2 expression on dsRNAi in F9 teratocarcinoma cell lines and include these data as Supplemental Figure 6. We find that LGP2 expression has no appreciable effect on dsRNAi in F9 cells. As this embryonic cell line lacks a competent IFN system, we suspect that other ISGs may work in concert with LGP2 to suppress dsRNAi. This is now discussed in the revised manuscript.

LGP2-deficient cell lines:

- *In order to assess the relevance of endogenous LGP2 expression for RNAi, evaluating LGP2-deficient cells would be of the utmost importance.*

o Is it possible to restore RNAi in differentiated cells if LGP2 is knocked out?

o In MEFs do LGP2^{-/-} show the same RNAi activity as IFNAR^{-/-}? Does the additional KO of LGP2 via CRISPR/Cas9 enhance what is already observed in IFNAR^{-/-} MEFs?

o Is the existing RNAi in cell lines such as P19 or F9 enhanced by LGP2 deficiency?

See above. We find that loss of LGP2 indeed restores dsRNAi, although less strongly than loss of the IFN receptor. We have included these data as a new Figure 7.

In the absence of an IFN response (e.g. in *Ifnar^{-/-}* MEFs or in the embryonic cell lines P19 and F9), there is no LGP2 expression, hence the knockout of LGP2 in such cell lines is uninformative.

Minor Points

Figure 5/6:

- *IFNAR^{-/-}MEFs without LGP2 but with doxycycline treatment should be included in both figures (or the supplementary figures).*

We have now included this control as Supplemental Figure 5A.

LGP2 Expression:

- *Has the correlation between basal LGP2 levels and RNAi activity been investigated? How much LGP2 is expressed in IFNAR^{-/-} MEFs?*

In the absence of IFN signaling, LGP2 is expressed at minimal levels in all cell types tested. We now include expression levels of LGP2 prior and post IFN treatment in wild type MEFs as Supplemental Figure 4A.

References

- Billy, E., Brondani, V., Zhang, H., Müller, U., and Filipowicz, W. (2001). Specific interference with gene expression induced by long, double-stranded RNA in mouse embryonal teratocarcinoma cell lines. *Pnas* 98, 14428-14433.
- Rothenfusser, S., Goutagny, N., DiPerna, G., Gong, M., Monks, B.G., Schoenemeyer, A., Yamamoto, M., Akira, S., and Fitzgerald, K.A. (2005). The RNA helicase *Lgp2* inhibits TLR-independent sensing of viral replication by retinoic acid-inducible gene-I. *The Journal of Immunology* 175, 5260-5268.
- Satoh, T., Kato, H., Kumagai, Y., Yoneyama, M., Sato, S., Matsushita, K., Tsujimura, T., Fujita, T., Akira, S., and Takeuchi, O. (2010). LGP2 is a positive regulator of RIG-I- and MDA5-mediated antiviral responses. *Proc. Natl. Acad. Sci. U.S.A.* 107, 1512-1517.

2nd Editorial Decision

30 November 2017

Thank you for submitting your manuscript to The EMBO journal. Your revision has now been re-reviewed by the three referees. As you can see below the referees appreciate the introduced changes and support publication in the EMBO Journal. I am therefore very pleased to accept the manuscript for publication here.

As soon as I get the revised version back in I will send formally accept the manuscript.

Congratulations on a great paper

REFEREE REPORTS

Referee #1:

I am satisfied by the answers that were made to the points I raised, and recommend publication of this important manuscript.

Referee #2:

In this revised manuscript, Van der Veen et al provide a role for LGP2 in inhibiting dsRNAi in mammalian cells. The authors have adequately addressed my experimental concerns where applicable; furthermore, the discussion has been improved to showcase the papers' findings in the appropriate context. Well done.

Referee #3:

The revised manuscript thoroughly addresses the critical questions raised by the reviewers. I have no further comments or concerns.

2nd Revision - authors' response

8 December 2017

In this version, we have addressed all reviewers' comments with additional data and revisions to the text as detailed in our point-by-point reply. I hope that the reviewers will find the manuscript improved and acceptable for publication. I take the opportunity to thank you for your support and look forward to hearing back from you in due course.

Corresponding Author Name: Caetano Reis e Sousa

Journal Submitted to: The EMBO Journal

Manuscript Number: EMBOJ-2017-97479